# Low-dose ethanol consumption inhibits neutrophil extracellular traps formation to alleviate rheumatoid arthritis

Lin Jin[1,2,3,5], Ziwei Zhang[1,2,3,5], Pin Pan[4,5], Yuchen Zhao[1,2,3], Mengqi Zhou[1,2,3], Lianghu Liu[1,2,3], Yuanfang Zhai[1,2,3], Han Wang[1,2,3], Li Xu[1,2,3], Dan Mei[1,2,3], Han Zhang[1,2,3], Yining Yang[1,2,3], Jinghan Hua[1,2,3], Xianzheng Zhang [ID] [1,2,3 ✉] & Lingling Zhang [ID] [1,2,3 ✉]

Rheumatoid arthritis (RA) is a chronic systemic autoimmune disease. Ethanol consumption has been reported to reduce morbidity in RA patients, but the mechanism behind it remains unclear. Our results showed that *Muribaculaceae* was predominant in the gut microbiota of mice after ethanol treatment, and the levels of microbiota metabolite acetate were increased. Acetate reduced arthritis severity in collagen-induced arthritis (CIA) mice, which was associated with a decrease in the articular neutrophils and the myeloperoxidase-deoxyribonucleic acid complex in serum. Meanwhile, in vitro experiments confirmed that acetate affected neutrophil activity by acting on G-protein-coupled receptor 43, which reduced endoplasmic reticulum stress in neutrophils and inhibited neutrophil extracellular traps formation. Furthermore, exogenous acetate reversed CIA mice with exacerbated gut microbial disruption, further confirming that the effect of gut microbial metabolite acetate on neutrophils in vivo is crucial for the immune regulation. Our findings illuminate the metabolic and cellular mechanisms of the gut-joint axis in the regulation of autoimmune arthritis, and may offer alternative avenues to replicate or induce the joint-protective benefits of ethanol without associated detrimental effects.

[1] Institute of Clinical Pharmacology, Anhui Medical University, Hefei 230032 Anhui, China. [2] Key Laboratory of Anti-inflammatory and Immune Medicine, Ministry of Education, Hefei 230032 Anhui, China. [3] Anti-inflammatory Immune Drugs Collaborative Innovation Center, Hefei 230032 Anhui, China. [4] Department of orthopedics, The Second People's Hospital of Hefei, Hefei Hospital Affiliated to Anhui Medical University, Hefei 230011 Anhui, China. [5] These authors contributed equally: Lin Jin, Ziwei Zhang, Pin Pan. ✉email: zxzhang0514@163.com; ll-zhang@hotmail.com

Rheumatoid arthritis (RA) is a chronic systemic auto-immune disease with the presentation of erosive arthritis, and its pathological basis is synovitis[1]. The early manifestations of RA include morning stiffness, swelling, and pain of joints. In severe cases, joint deformity may occur and normal joint function may be lost[2]. The pathogenesis of RA is complex, and the interaction between genetic and environmental factors can affect the immunopathology of RA[3]. Ethanol ($C_2H_5OH$, EtOH; commonly referred to as alcohol) as part of the dietary composition has been shown to have effects on innate and adaptive immunity. In fact, moderate alcohol consumption has long been recognized as a protective factor in the pathogenesis of autoimmune diseases such as RA[4,5], but the specific mechanism of action has been unclear.

Gut microbiota can produce various metabolites, which enter the blood circulation and profoundly affect the human immune system[6]. More and more studies support that the manipulation of the gut microbiota may become a new strategy for the prevention and/or treatment of RA[7–9]. Short-chain fatty acids (SCFAs), which are organic fatty acids with 1–6 carbons, are the major products of bacterial fermentation of carbohydrates such as dietary fiber and resistant starch. It has been reported that low-dose ethanol could be metabolized by the gut microbiota into SCFAs, in which the ratio of acetic acid, propionic acid, and butyric acid was about 60:20:20[6]. SCFAs can bind to receptors on innate immune cells, such as free fatty acid receptor 3 (GPR41), free fatty acid receptor 2 (GPR43) and G-protein coupled receptor 109A (GPR109A) highly expressed on the surface of innate immune cells represented by neutrophils[10]. They play a key role in regulating immunity, improving gut barrier function, inhibiting tumor growth, and regulating gene expression in various diseases[10]. However, the effects of SCFAs on neutrophils and RA have been rarely reported.

Neutrophils are the most abundant white blood cells in peripheral blood. As innate immune cells, neutrophils play an important role in the early stage of RA and are involved in the regulation of immunopathology of RA[11]. Neutrophil extracellular traps (NETs) are the unique weapons of neutrophils to clear pathogens. When neutrophils are stimulated by autoantibodies or other exogenous stimuli, the neutrophil elastase and other serine proteases regulated by myeloperoxidase (MPO) will be released. These proteases will then translocate protein arginine deiminase 4 to the nucleus, which induces the citrullination of histone arginine residues to produce citrullinated histone H3 (H3cit) and promotes chromatin dissociation and condensation, resulting in the destruction of the neutrophil nuclear membrane. The decompressed chromatin is released into the cytoplasm and further modified by dozens of proteases such as MPO, neutrophil elastase, and cathepsin G[12–14]. The protease-modified chromatin passes through the damaged cell membrane and is released to the outer space of the cell to form a "trap net". This unique generation process of NETs is called "NETosis"[15,16].

Although NETs are important host defense mechanisms[17], their abnormal formation and clearance can lead to tissue damage and abnormal activation of the immune system in autoimmune diseases such as RA[18]. It has been reported that excessive activation of the Endoplasmic reticulum stress (ERS) response sensor inositol-requiring enzyme 1 (IRE1) in neutrophils can further promote the generation of NETs and aggravate disease progression[19]. ERS is a reaction induced by the accumulation of a large amount of unfolded or misfolded proteins in the lumen of the endoplasmic reticulum. To resolve ERS and protect cells, an unfolded protein response will be triggered, and the expression of endoplasmic reticulum chaperones such as glucose-regulated protein (GRP78) and cell apoptosis will be induced. The unfolded protein response primarily involves three ERS response sensors,

including IRE1, the activating transcription factor 6 (ATF6), and protein kinase RNA–like endoplasmic reticulum kinase. It is unclear whether ERS in neutrophils is associated with the development of RA.

In this study, we first detected arthritis indicators in a low-dose alcohol treatment model of CIA mice and found that ethanol treatment can alleviate the progression of CIA mice. Using a range of Tissue staining observation and biological detection technology, we elucidated the ethanol treatment inhibits the formation of NETs in the joints. Then, microbiomic and metabolomic analyses revealed that *Muribaculaceae* was predominant in the gut microbiota of mice after ethanol treatment, and the levels of microbiota metabolite acetate were increased. Meanwhile, acetate reduced ERS in neutrophils and inhibited the formation of NETs. Furthermore, exogenous acetate reversed CIA mice with exacerbated gut microbial disruption, further confirming that the effect of gut microbial metabolite acetate on neutrophils in vivo is crucial for the immune regulation. This work emphasizes the pivotal role of acetate during RA intervention, which may offer alternative avenues to replicate or induce the joint-protective benefits of ethanol without associated detrimental effects.

## Results

**Low-dose ethanol treatment alleviates CIA in mice.** To evaluate the effect of low-dose ethanol treatment on CIA, mice were randomly divided into the control (NC group), CIA model (CIA group), ethanol-only feeding (EtOH group), and ethanol-feeding CIA (EtOH + CIA group) groups. As shown in Fig. 1a, after acclimatization, DBA/1 mice were immunized with collagen II on day 0 and day 21 to induce CIA. Mice in the ethanol-feeding CIA group were given adaptive water feeding containing ethanol before immunization, and drinking water containing 10% (v/v) ethanol was given from day 0 after the first immunization. The development of arthritis in CIA mice was monitored from the second collagen immunization under normal feeding and low-dose ethanol treatment (Supplementary Fig. 1a). Paw inflammation in mice peaked around day 36 after the first immunization, and there were evident redness and swelling in the ankles, front paws, and hind paws of CIA mice (Fig. 1b and Supplementary Fig. 1b). Compared with the CIA model group, the arthritis symptoms of mice in the ethanol-feeding CIA group were reduced; the incidence of CIA was reduced; the disease progression was slowed down (Fig. 1); and the clinical manifestations were alleviated (Fig. 1d). As revealed by histopathological staining, 10% (v/v) ethanol treatment did not cause obvious changes in the liver of mice (Supplementary Fig. 1c), and there was no obvious sign of arthritis in the mouse joints (Supplementary Fig. 1d–g), indicating that consumption of low-dose ethanol may not have adverse/toxic effects on mice.

Compared with the control group, the spleen of the CIA mice was enlarged with dark red color, and the spleen coefficient was increased. However, the spleen coefficient of mice in the ethanol-feeding CIA group was lower than that of the CIA model group (Fig. 1e). Germinal centers were massively increased during immune abnormalities, but the number of germinal centers was decreased in the spleen of CIA mice treated with ethanol (Fig. 1f). Ultrasonic examination of the knee joints of CIA mice revealed a large amount of anechoic fluid in the joint cavity of CIA mice, increased joint effusion, and synovial hyperplasia (white arrow). Color Doppler flow imaging showed that the blood flow signal in the joints was evidently weakened after ethanol treatment (yellow arrow), and the synovitis was alleviated (Fig. 1g). X-ray imaging and histopathological staining were used to evaluate the effect of ethanol therapy on the progression of arthritis. Normal joints are

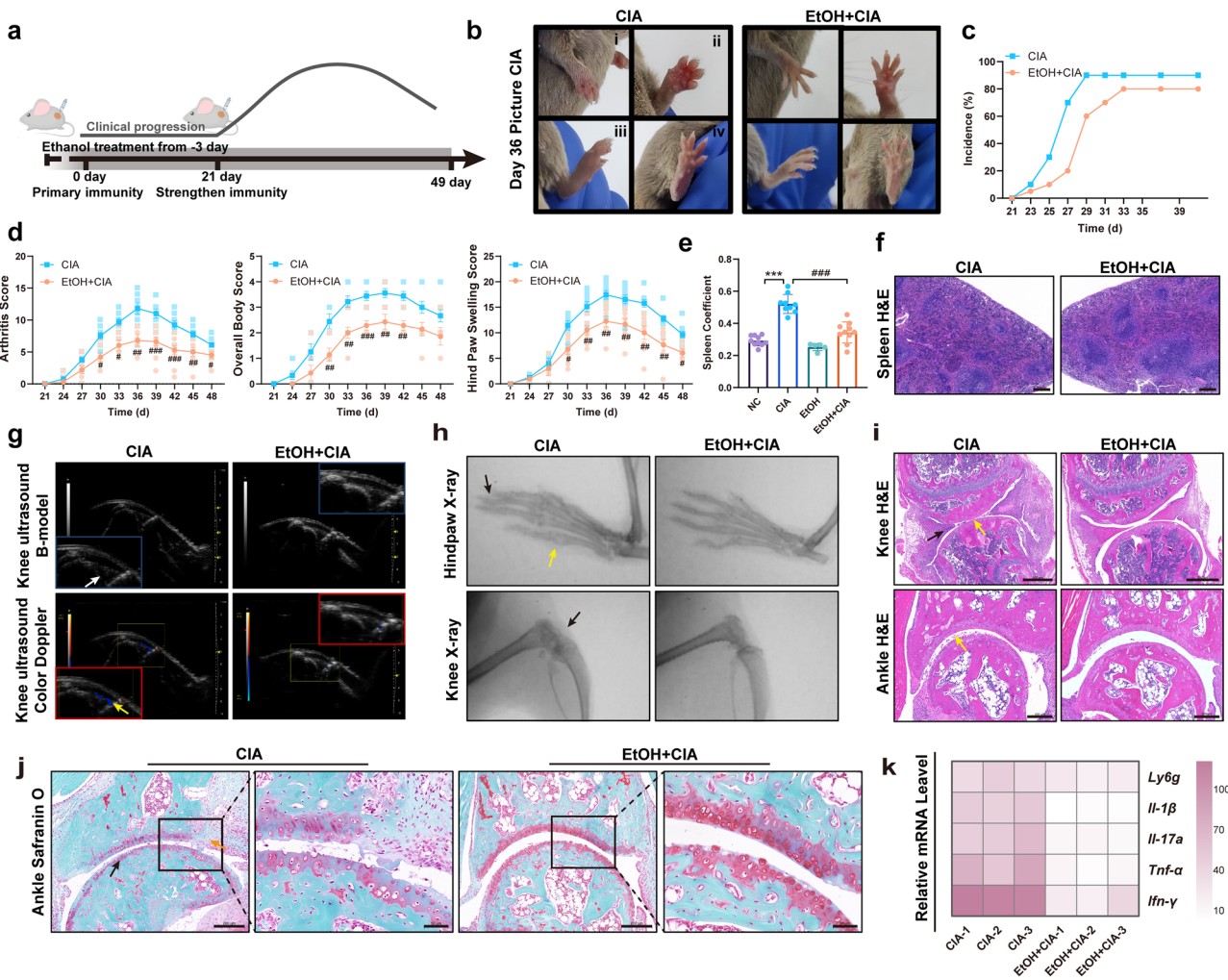

**Fig. 1 Low-dose ethanol treatment attenuates clinical manifestations of CIA mice. a** Flowchart of CIA mouse model establishment and ethanol treatment. **b** Photographs of the hind paws and front paws on day 36 after the first immunization (i, the front of the front paw; ii, the back of the front paw; iii, the front of the hind paw; iv, the back of the hind paw). **c** Incidence of CIA, defined as Hind Paw Swelling Score > 0 per mouse. **d** Arthritis severity during CIA, as assessed by arthritis score, overall body score, and hind paw swelling score. #$p < 0.05$, ##$p < 0.01$, and ###$p < 0.001$ represent significant differences between the CIA group and the EtOH + CIA group (CIA $n = 9$, EtOH + CIA $n = 9$). Data are presented as mean ± SEM and analyzed with Two-way ANOVA followed by Sidak's post hoc test. **e** Statistical analysis of spleen coefficient. ***$p < 0.001$ versus the control group and ###$p < 0.001$ versus the CIA group (NC $n = 9$, CIA $n = 8$, EtOH $n = 5$, EtOH + CIA $n = 9$). Data are described as mean ± SEM and compared with One-way ANOVA followed by Tukey's post hoc test. **f** H&E images of spleen sections (Scale Bar: 200 μm). **g** Image of knee joints of mice on ultrasonography. Longitudinal B-mode imaging shows a large amount of anechoic fluid in the joint cavity, increased joint effusion, and synovial hyperplasia (white arrow). Color Doppler flow imaging shows the blood flow signal in the joint (yellow arrow). **h** X-ray images of joints of mice show macroscopic evidence of arthritis. Swelling or scattered toe bones and genu bones (black arrow), and markedly narrowed joint space (yellow arrow). **i** H&E images show the dominant histological structure of the knee joints. Synovial intima hyperplasia (black arrow) and inflammatory cell infiltration (yellow arrow) (Scale Bar: 500 μm and 200 μm). **j** Representative photomicrographs of joint histopathology after staining with safranin O-fast green. Histopathological changes including cartilage and bone damage (black arrows) and inflammatory cell infiltration (yellow arrows) (Scale Bar: 200 μm and 50 μm). **k** The mRNA expressions of related cytokines in mouse paw by RT-qPCR.

intact, the joint space is clear and smooth, and the destruction of bone and cartilage is one of the main pathological manifestations of RA. On X-ray imaging, the macroscopical evidence of arthritis was evident in the CIA mice, such as blurring of ankle and toe joints (black arrow), narrowing of joint space (yellow arrow), cystic changes, and apparent erosion and degradation of bones. Synovial intima hyperplasia (black arrow) and inflammatory cell infiltration (yellow arrow) are observed in CIA mice (Fig. 1h). Of note, ethanol treatment reduced the severity of arthritis in CIA mice. Histological analysis of the mouse knee and ankle joints showed that compared with the CIA model group, the synovial inflammation in the ethanol-feeding CIA group was reduced (Fig. 1i). Furthermore, there were decreases in bone erosion

(black arrows) and inflammatory cell infiltration (yellow arrows) in ethanol-treated CIA mice (Fig. 1j). RT-qPCR analysis of whole paw homogenates showed that ethanol treatment reduced the mRNA levels of *Il-1β*, *Il-17a*, *Ifn-γ*, and *Tnf-α*, which are key inflammatory cytokines in the joints of CIA mice[20], and the mRNA level of neutrophil marker *Ly6g* (Fig. 1k). These data suggest that low-dose ethanol consumption alleviates the clinical manifestations of CIA mice and protects bone from erosion.

**Ethanol treatment inhibits the infiltration of neutrophils and the formation of NETs in the joints of CIA mice.** The proportion of immune cells in the paws of mice was detected by flow cytometry. The results showed that a variety of immune cells were

detected in the paws of CIA mice, while ethanol treatment reduced the infiltration of immune cells. In detail, the white blood cells (CD45[+]), T cells (CD3[+]), monocytes and macrophages (CD11b[+]Ly6C[+]) were reduced, while the decrease of neutrophils (CD11b[+]Ly6G[+]) was the most significant ($p = 0.0009$) (Fig. 2a–e and Supplementary Fig. 2a–c), which was consistent with the decrease of *Ly6g* gene expression. ELISA found that after ethanol treatment, the level of MPO-DNA, a marker of NETs, in the serum of CIA mice was reduced (Fig. 2f). This data suggests that ethanol treatment may further alleviate CIA progression by affecting the formation of NETs. In addition, immunohistochemistry and immunofluorescence confirmed that NETs in the ankle joints of CIA mice were reduced after ethanol treatment (Fig. 2g, h). Taken together, these data demonstrated that ethanol treatment can effectively reduce the infiltration of various immune cells, with the most significant effect on neutrophils, and inhibit the formation of NETs in the joints during the pathogenesis of arthritis.

**The reduced production of NETs is associated with the inhibition of ERS**. It has been shown that over-activation of the ERS sensor IRE1 in neutrophils can further promote the release of NETs, which is closely related to disease severity in lupus[19]. When ERS occurs, IRE1 is activated through autophosphorylation, showing endonuclease activity, cutting the downstream molecule (XBP1) base of IRE1 signaling pathway, and generating an active spliceosome XBP1s to regulate the downstream. However, the role of ERS in the pathogenesis of autoimmune arthritis remains unclear. Our results found that ethanol treatment reduced the phosphorylation of IRE1 (p-IRE1) in neutrophils while reducing the formation of NETs (Fig. 3a). Therefore, the phosphorylation of IRE after involvement in ERS may be involved in autoimmune arthritis by promoting the formation of NETs.

This assumption was verified by subsequent experiments on RA patients. Immunohistochemical results showed that there were a large number of neutrophils in the arthritic synovial tissues of RA patients compared with osteoarthritis (OA) patients (Fig. 3b). The neutrophils of healthy volunteers (HC) and RA patients were isolated from the peripheral blood. Western blot results showed that the expressions of GRP78, p-IRE1, XBP1, H3cit, and MPO were increased in neutrophils from RA patients (Fig. 3c), which indicates that there is activation of ERS response sensor IRE1 phosphorylation and increased production of NETs in RA. In addition, immunofluorescence staining also confirmed that there was over-production of p-IRE1 in the neutrophils of synovial tissues of RA patients (Fig. 3d). Therefore, the reduction of NETs may be related to the suppression of ERS in neutrophils.

**Ethanol treatment alters the gut microbiota in CIA mice**. It is shown that there is an imbalance of gut microbiota in RA patients, and there are also differences in the gut microbiota of RA patients before and after treatment. Gut microbiota is closely related to the occurrence, development, and even treatment of RA. 16S rRNA sequencing was used to examine the changes in the gut microbiota of CIA mice and the effect of ethanol treatment on these changes. As shown in Fig. 4a, there were no significant difference among the diversities of gut microbiota in mice of the control group, CIA group, EtOH group, and EtOH + CIA group. Compared with the control and CIA groups, there were 42 fewer shared operational taxonomic units (OTUs) between the control and EtOH + CIA groups (Fig. 4b), suggesting that ethanol treatment may reduce the similarity of gut microbiota between control mice and CIA mice, and improve the gut microbiota of CIA mice. There were 312, 315, 280, and 220 independent OTUs

in the control, CIA, EtOH, and EtOH + CIA groups, respectively. Principal component analysis of OUTs was used to measure changes in the gut microbial composition of CIA and EtOH-treated mice, and based on the Bray–Curtis differences in OTUs in the gut microbiota of mice in each group, the β diversity of gut microbiota was compared by weighted unified principal coordinate analysis. Compared with control and CIA mice, a large number of specific clusters was generated in mice after ethanol treatment (Fig. 4c, d).

According to the relative abundance of gut microbiota at the family level, *Muribaculaceae*, *Lactobacillaceae*, and *Lachnospiraceae* were found to be different among the groups. Compared with CIA mice, the increase of *Muribaculaceae* was most obvious after ethanol treatment. *Muribaculaceae*, a genus producing SCFAs, mainly produced acetate and propionate as the main fermentation substrates (Fig. 4e). The relative abundance of gut microbiota at the genus level in each group was analyzed by heat map analysis (Fig. 4f), and the number of beneficial gut microbiota in CIA mice increased after ethanol treatment, such as *Clostridia* and *Muribaculum*. Conditional pathogenic bacteria such as *Desulfovibrio* and *Roseobacter*, which fully utilize acetic acid, lactic acid, and fatty acid as carbon and energy sources, were reduced. Linear discriminant analysis indicated differences in the genus-level relative abundances of gut microbiota among the NC, CIA, EtOH, and EtOH + CIA groups (Fig. 4g), consistent with the heat map analysis. Collectively, ethanol treatment can alter the gut microbiota of CIA mice.

**Ethanol treatment increases the production of microbiota metabolite acetate**. Bacteroides can efficiently ferment the substrates in the gut into SCFAs, which have anti-inflammatory and immunomodulatory effects. Since there was a higher abundance of strains in the gut microbiota of ethanol-treated CIA mice, we hypothesize that SCFAs might play a protective role in CIA mice. To this end, GC-MS analysis was performed to quantify SCFAs in serum. As shown in Fig. 5a, b, acetate level was increased in ethanol-treated CIA mice. SCFAs may exert their effects by activating G protein-coupled receptors (GPCRs). Therefore, the mRNA expression levels of free fatty acid receptors (FFARs), including GPR41, GPR43 and GPR109A, in the colon were further quantified. Surprisingly, the *Gpr43* mRNA level was increased in mice treated with ethanol, while the *Gpr41* and *Gpr109a* mRNA levels were not significantly affected (Fig. 5c), which further confirms that gut microbiota-derived acetate may play a key protective role in the progression of related diseases.

**Acetate activates GPR43 and inhibits ERS to reduce the formation of NETs**. The intake of SCFAs is one of the factors affecting the occurrence and development of RA, and SCFAs such as acetic acid and propionic acid have protective effects[21,22]. However, the underlying mechanism remains unclear. It has been reported that low-dose ethanol can be metabolized to SCFAs by gut microbiota, in which acetate has the highest proportion[6]. In vivo experiments showed that ethanol treatment did alter the gut microbiota and the production of SCFAs in CIA mice, with the most significant change in the production of acetate. In vitro experiments were carried out to observe the specific effect and mechanism of acetate on neutrophils and to clarify the specific mechanism of acetate in alleviating CIA mice. Cell morphology analysis revealed that acetate inhibited the formation of NETs. On scanning electron microscopy, NETs were observed after 4 h of in vitro treatment of neutrophils with PMA (Fig. 6a). However, under the co-stimulation of acetate and PMA, the original cell morphology barely changed and there was no diffusion. Multiple frills were also observed on the cell surface. NETs can be

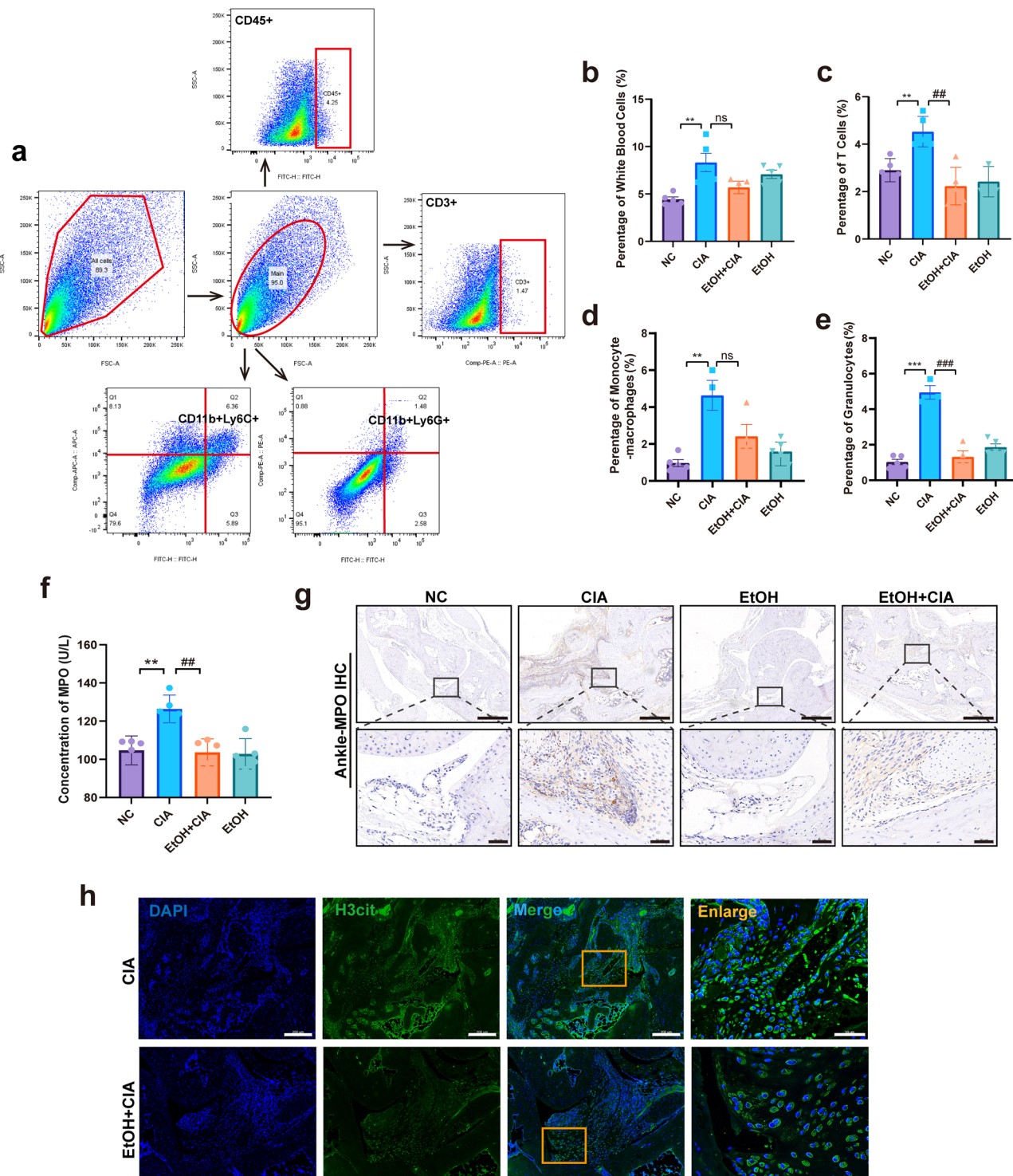

**Fig. 2 Ethanol treatment inhibits the formation of NETs in the joints of CIA mice. a** The gating strategy for flow cytometry experiments. **b–e** The percentage of white blood cells (CD45$^+$), T cells (CD3$^+$), monocytes and macrophages (CD11b$^+$Ly6C$^+$), and neutrophils (CD11b$^+$Ly6G$^+$) in the total number of cells in the paws of each group of mice. *$p < 0.05$, **$p < 0.01$, and ***$p < 0.001$ versus the control group; #$p < 0.05$, ##$p < 0.01$, and ###$p < 0.001$ versus the CIA group ($n = 5$ per group); and, ns no significance. Data are presented as mean ± SEM and compared with One-way ANOVA followed by Tukey's post hoc test. **f** The level of MPO-DNA in serum from different groups. *$p < 0.05$ and **$p < 0.01$ versus the control group; #$p < 0.05$ and ##$p < 0.01$ versus the CIA group ($n = 5$ per group). Data are presented as mean ± SEM and compared with One-way ANOVA followed by Tukey's post hoc test. **g** Immunohistochemical staining of MPO in mouse ankle joints (Scale Bar: 500 µm and 50 µm). **h** The formation of NETs in mouse ankle joints was observed by immunofluorescence staining of H3cit (green) and DAPI (blue) (Scale Bar: 200 µm and 50 µm).

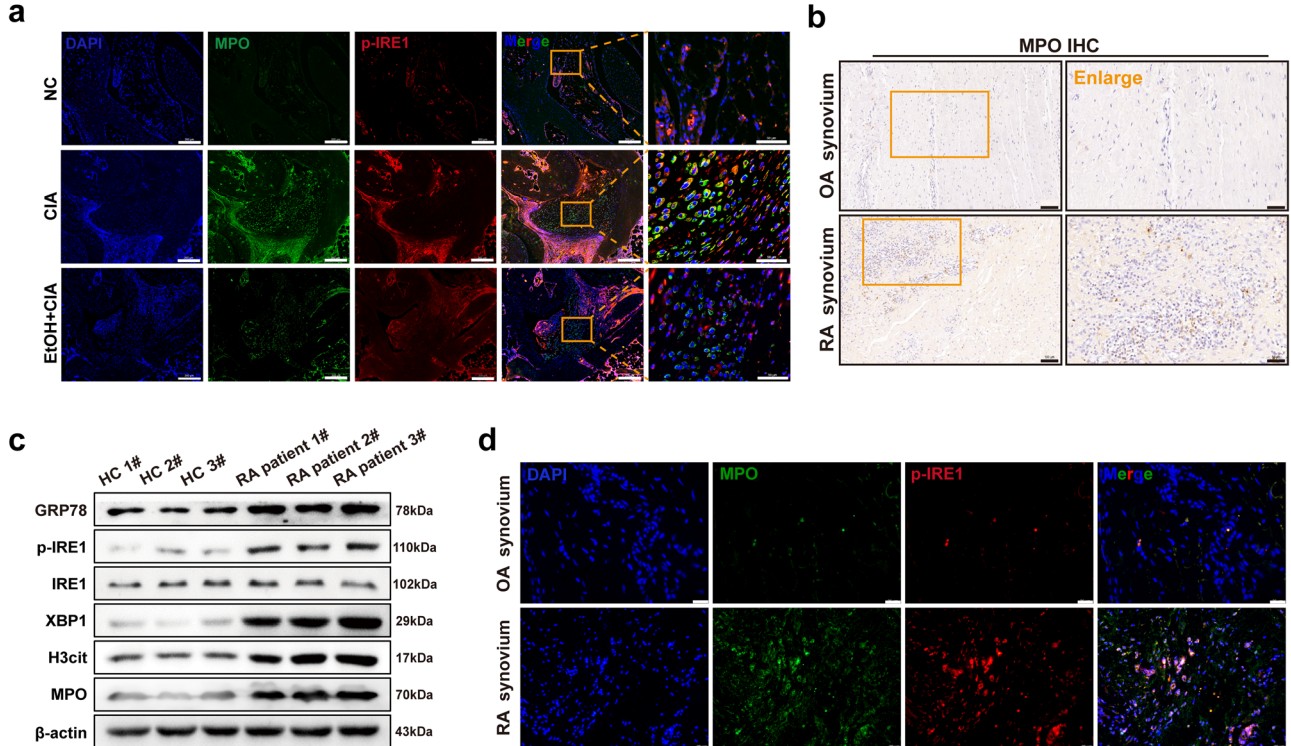

**Fig. 3 Ethanol treatment may be associated with the inhibition of ERS and upregulation of IRE1-NETs. a** Immunofluorescence staining of MPO (green) and p-IRE1 (red) in the ankle joints of mice. MPO and p-IRE1, which indicate NETs and ERS, respectively, are co-localized in the ankle joints of CIA mice (Scale Bar: 200 μm and 50 μm). **b** Immunohistochemical staining of MPO in the joint synovium of patients with OA and RA (Scale Bar: 100 μm and 50 μm). **c** Western blot detection of neutrophils in peripheral blood of healthy volunteers and RA patients. Data are representative of three independent experiments. **d** Immunofluorescence staining of MPO (green) and p-IRE1 (red) in the synovium of OA and RA patients. ERS is observed in neutrophils of the synovium of RA patients (Scale Bar: 25 μm).

identified by staining extranuclear DNA with Sytox-green. The results of immunofluorescence staining showed that acetate inhibited the formation of NETs (Fig. 6b), which was consistent with the results of electron microscopy. Live cell confocal microscopy observed the production of NETs within four hours, and it was found that the formation of NETs was delayed and inhibited under the treatment of acetate (Supplementary Videos 1a, 1b). Moreover, acetate treatment reduced MPO and p-IRE1 co-localization in neutrophils (Fig. 6c). In addition, acetate reduced the mRNA expression of inflammatory cytokines, including *Il-1β*, *Il-17a*, *Cxcl1*, *Cxcl2*, and *Ccl2*, as revealed by RT-qPCR (Supplementary Fig. 3a).

Neutrophils promote NETs release in an IRE1-dependent manner[19]. To further investigate the effects of acetate on neutrophils, Western blot analysis was performed on neutrophils stimulated by different concentrations of acetate and PMA. The results showed that ERS response sensor IRE1 was inhibited and p-IRE1 expression was decreased with the increase of acetate concentration, while there was no significant change in the expression of CHOP and ATF6. Thus, acetate may reduce the release of NETs by inhibiting the sensor IRE1 in a dose-dependent manner, and the optimal inhibitory effect of acetate was achieved at 10 mM. Meanwhile, the expressions of p-IRE1 and XBP1 were the lowest, and the expressions of H3cit and MPO were all reduced (Fig. 6d). It was acetate rather than ethanol itself that produced this inhibitory effect (Fig. 6e). In addition, Western blot analysis showed that thapsigargin (TG), an ERS inducer, activated neutrophils to promote the release of NETs in an IRE1-dependent manner, and after treatment with acetate, the expressions of GRP78, IRE1, XBP1, H3cit, and MPO were reduced (Fig. 6g), suggesting that acetate inhibits the formation of

NETs by reducing neutrophil p-IRE1/XBP1. And the expression of GPR43 was increased in the presence of acetate (Fig. 6f). GPR43, which is abundantly expressed in immune cells such as neutrophils, can be activated by SCFAs and plays an important role in regulating physiological processes such as white blood cell function in the body and the nutrient absorption in the intestinal tract of mammals[23–25]. Western blot results showed that acetate may reduce ERS in neutrophils to inhibit the formation of NETs by promoting the phosphorylation of G protein-coupled receptor kinase 2 (GRK2) at S670 in neutrophils (Fig. 6f). Therefore, a possible mechanism is that acetate promotes the phosphorylation of GRK2 at S670 by acting on GPR43, thereby promoting the ubiquitination and degradation of GRK2. Furthermore, this interaction was confirmed by immunoprecipitation, and ubiquitination was involved in the regulation of GRK2 expression under the treatment of acetate (Fig. 6h). Overall, the observed inhibition of NETs by acetate could be attributed to the fact that acetate acts on GPR43 to inhibit p-IRE1/XBP1 and reduce NETs formation in neutrophils.

**Exogenous supplementation of acetate alleviates inflammation in CIA mice.** The protective effect of acetate on CIA was further investigated. According to different treatments, the mice were randomly divided into the CIA model, CIA + ABX, CIA + ACE, and CIA + ABX + ACE groups. After acclimatization, DBA/1 mice were subjected to the first immunization on day 0, and the gut microbiota of mice in the CIA + ABX group were continuously disrupted with antibiotics (Fig. 7a). Mice in the CIA + ACE group were given drinking water containing acetate to supplement acetate from day 0. After the second immunization, the clinical manifestations of CIA mice were scored every 2

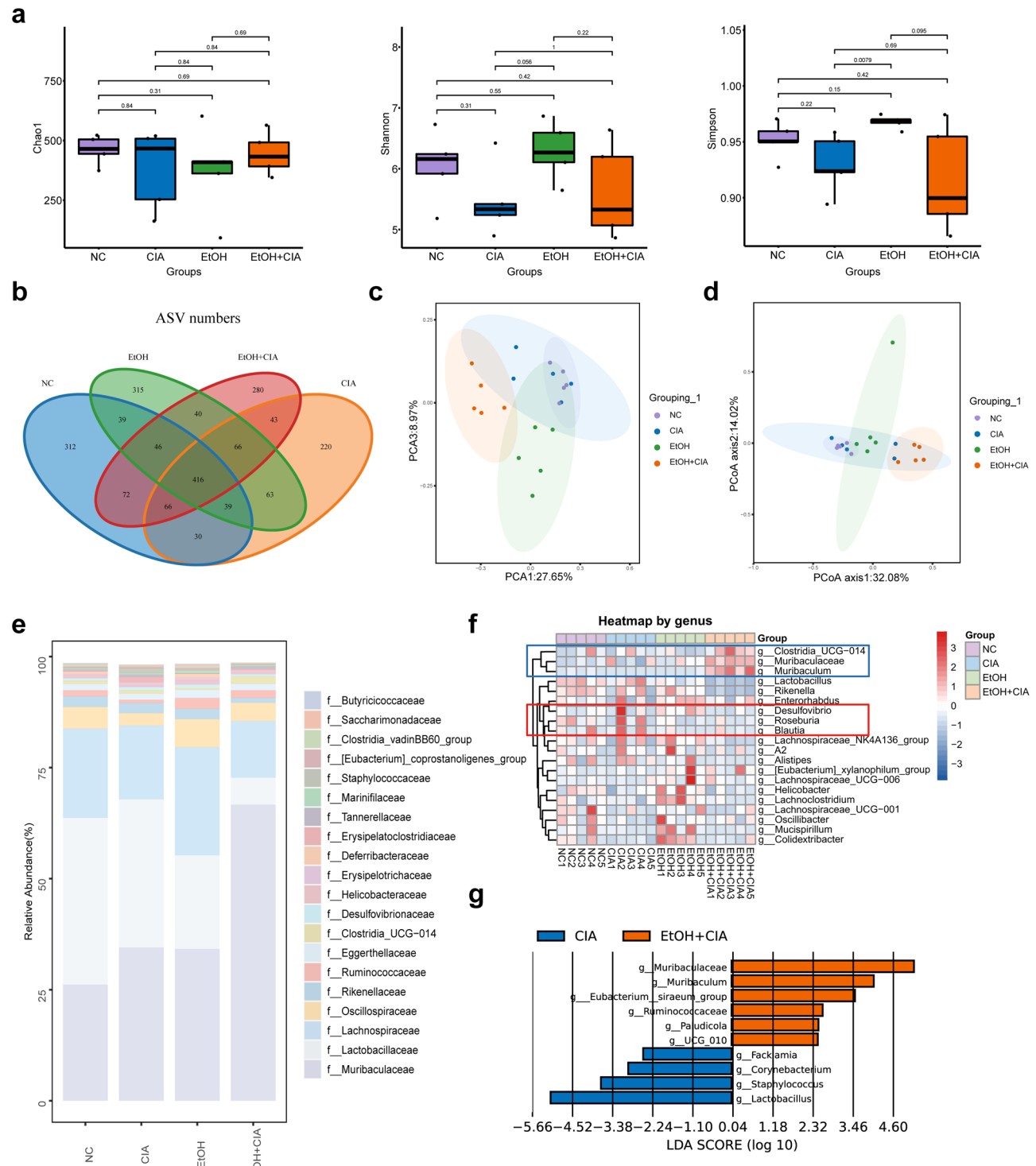

**Fig. 4 Ethanol alters the gut microbiota of CIA mice. a** The diversity of microbial species in NC, CIA, EtOH, and EtOH + CIA mice ($n = 5$ per group) are evaluated by the Chao index, Shannon index, and Simpson index, respectively. **b** Venn diagram of the overlap of bacterial operational taxonomic unit (OTU) among mice of NC, CIA, EtOH, and EtOH + CIA groups. **c** Changes in the gut microbial composition of CIA and EtOH-treated mice measured by principal component analysis. **d** Comparison of gut bacterial β-diversity by weighted unified principal coordinates analysis based on Bray–Curtis differences in OTUs in the gut microbiota of mice in each group. **e** Relative abundance of gut bacterial families in each group. **f** Relative abundance of gut microbial genera in each group is analyzed by heat map analysis, and the sample clustering shows the abundance of important bacterial groups in different groups. **g** Linear discriminant analysis indicating significant differences in the abundance of gut bacterial genera between the CIA and EtOH + CIA or NC, CIA, EtOH, and EtOH + CIA groups. High linear discriminant analysis scores indicate that the species abundance has a strong effect on the differences between groups.

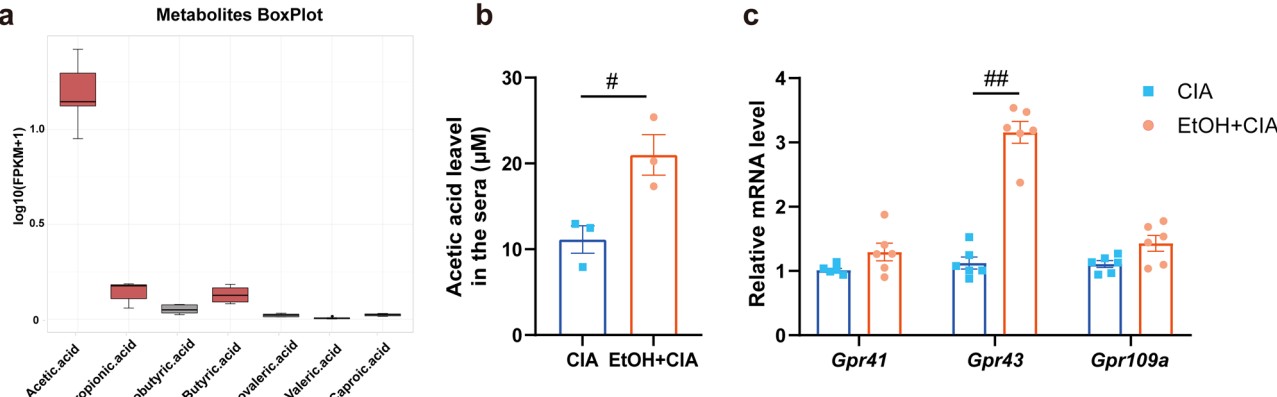

**Fig. 5 Microbiota metabolite acetate production increases in ethanol-treated mice. a, b** Concentrations of SCFAs in the serum of mice in the CIA and EtOH + CIA groups ($n = 3$ per group). #$p < 0.05$ versus the CIA group. Data are presented as mean ± SEM and compared with unpaired $t$ tests. **c** The mRNA expressions of FFARs, including *Gpr41*, *Gpr43*, and *Gpr109a*, in the gut. #$p < 0.05$ versus the CIA group ($n = 6$ per group). Data are presented as mean ± SEM and compared with Two-way ANOVA followed by Sidak's post hoc test.

**Fig. 6 Acetate reduces ERS in neutrophils and inhibits the formation of NETs. a** Scanning electron microscopy observation of treated neutrophils (Scale Bar: 2 μm). i: Neutrophil-like differentiation of HL-60 (dHL-60) cells that developed NETs after 4 h of PMA treatment in vitro. ii: There were no NETs in dHL-60 cells treated with acetate and PMA. **b** Immunofluorescence staining of Sytox-green (green) and MPO (red) in dHL-60 cells, showing that NETs are reduced under sodium acetate treatment (Scale Bar: 50 μm and 25 μm). **c** Immunofluorescence staining of MPO (green) and p-IRE1 (red) in dHL-60 cells, showing that MPO and p-IRE1 co-localization is reduced under sodium acetate treatment (Scale Bar: 10 μm and 50 μm). **d** The effects of GRP78/p-IRE1/IRE1/CHOP/ATF6/XBP1/H3cit/MPO in dHL-60 cells exposed to different concentrations of acetate. Data are representative of three independent experiments. **e** The effects of H3cit/MPO in dHL-60 cells exposed to different concentrations of ethanol. Data are representative of three independent experiments. **f** The effects of acetate receptor GPR43 and its downstream proteins in dHL-60 cells after stimulated by TG. Data are representative of three independent experiments. **g** The effects of GRP78/p-IRE1/IRE1/CHOP/ATF6/XBP1/H3cit/MPO in dHL-60 cells after stimulated by TG. Data are representative of three independent experiments. **h** Immunoprecipitation analysis of ubiquitination of endogenous GRK2 in dHL-60 cells. Data are representative of three independent experiments.

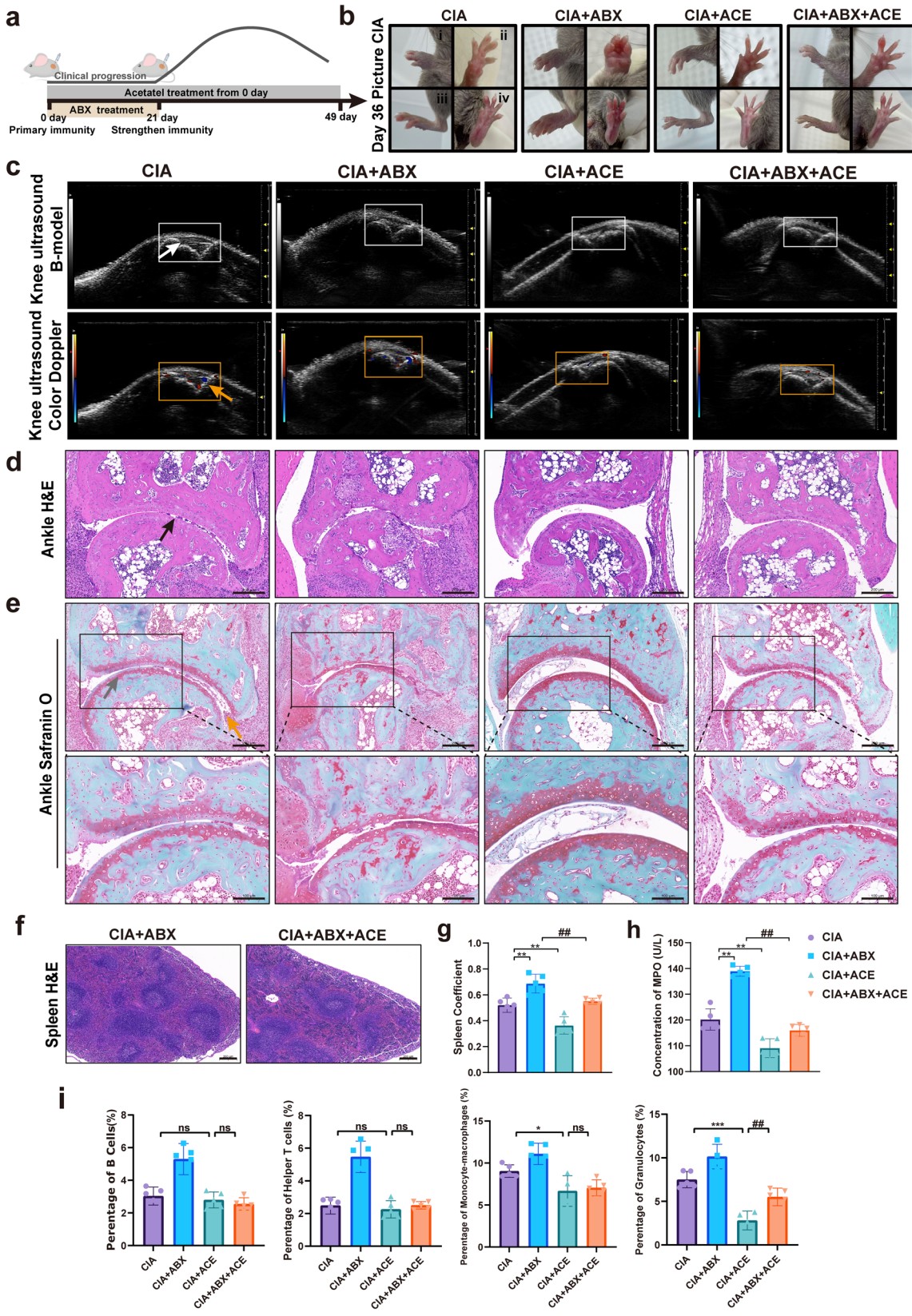

days. Paw inflammation in mice peaked around 36 days after immunization, and acetate supplementation induced a decrease in CIA mice (Fig. 7b). Antibiotics disrupted the gut microbiota of CIA mice and aggravated the disease symptoms, as evidenced by severe redness and swelling of ankles and paws. Compared with the mice in the CIA + ABX group, the clinical scores of the mice in the CIA + ABX + ACE group were reduced (Supplementary Fig. 4a, b). The imaging results showed that compared with the

**Fig. 7 Exogenous acetate supplementation has a therapeutic effect on CIA mice. a** Flowchart of CIA mouse model establishment and intervention.
**b** Photographs of the hind paws and front paws on day 36 after the first collagen II immunization (i, the front of the front paw, ii, the back of the front paw, iii, the front of the hind paw, d, the back of the hind paw). **c** Image of knee joints of CIA mice on ultrasonography. Longitudinal B-mode imaging shows a large amount of anechoic fluid in the joint cavity, increased joint effusion, and synovial hyperplasia (white arrow). Color Doppler flow imaging shows the blood flow signal in the joint (yellow arrow). **d, e** Representative photomicrographs of H&E (upper) and safranin O-fast green staining (lower) of knee joints (Scale Bar: 200 μm and 100 μm). **f** H&E images of spleen sections (Scale Bar: 200 μm). Histopathological changes including cartilage and bone damage (gray arrows) and inflammatory cell infiltration (yellow arrows). **g** Statistical analysis of spleen coefficient. $*p < 0.05$ and $**p < 0.01$ versus the CIA group; $\#p < 0.05$ and $\#\#p < 0.01$ versus the CIA + ABX group ($n = 5$ per group). Data are presented as mean ± SEM and compared with Two-way ANOVA followed by Sidak's post hoc test. **h** The level of MPO-DNA in serum detected by ELISA. $*p < 0.05$ and $**p < 0.01$ versus the CIA group; $\#p < 0.05$ and $\#\#p < 0.01$ versus the CIA + ABX group ($n = 5$ per group). Data are presented as mean ± SEM and compared with Two-way ANOVA followed by Sidak's post hoc test. **i** The percentage of B cells (CD45+CD19+), mature T lymphocytes (CD3+CD4+), monocytes and macrophages (CD11b+Ly6C+), and neutrophil (CD11b+Ly6G+) in the total number of cells in the paws of each group of mice. $*p < 0.05$, $**p < 0.01$, and $***p < 0.001$ versus the CIA group; $\#p < 0.05$, $\#\#p < 0.01$, and $\#\#\#p < 0.001$ versus the CIA + ACE group ($n = 5$ per group); and, ns, no significance. Data are presented as mean ± SEM and compared with Two-way ANOVA followed by Sidak's post hoc test.

CIA model group, the CIA + ABX group had narrowed joint space and more blurring in the toe joints (black arrow) (Supplementary Fig. 4c). Moreover, there was a large amount of anechoic fluid in the joint cavity, increased joint effusion (white arrow), synovial hyperplasia, stronger blood flow signal, and aggravated pannus (yellow arrow) (Fig. 7c). The acetate supplementation in the CIA + ACE group had an obvious therapeutic effect on CIA mice, and that in the CIA + ABX + ACE group reversed the effects of ABX on CIA mice, as shown by improved cartilage and bone damage (gray arrow), attenuated synovial intimal hyperplasia (black arrow), and reduced inflammatory cell infiltration (yellow arrow) (Fig. 7d, e). These results confirm that the gut microbial metabolite acetate alleviates CIA mice.

Histopathological staining showed that the number of germinal centers in the spleen of mice in the CIA + ABX group increased, which decreased after acetate treatment in the CIA + ABX + ACE group. Compared with the CIA model group, the mice in the CIA + ABX group had enlarged spleens and a higher spleen coefficient, whereas these effects were reversed in the CIA + ABX + ACE group after acetate treatment (Fig. 7f, g). ELISA found that acetate treatment reduced the MPO-DNA level in the serum of mice in the CIA + ACE and CIA + ABX + ACE groups (Fig. 7h). Further analysis of cells in mouse paw suspensions by flow cytometry showed that acetate treatment affected a variety of immune cells, and the number of neutrophils showed the most significant reduction (Fig. 7i and Supplementary Fig. 4d). Taken together, the above results indicate that acetate produced by the gut microbiota of mice could attenuate the level of inflammation and alleviate the progression of CIA mice by decreasing the expression of MPO-DNA in neutrophils.

## Discussion

Ethanol is the main ingredient in alcoholic beverages. Heavy ethanol consumption is a common social and health problem and can cause toxicity to the central nervous system and liver[26]. On the other hand, previous epidemiological studies have shown that small to moderate ethanol treatment has an inhibitory effect on the probability and severity of RA[27,28]. However, the mechanisms underlying the mitigation effects of alcohol on RA have only been partially elucidated. Vugar Azizovl et al. reported that ethanol inhibited inflammatory cytokines in vivo, thereby altering the activity of T follicular helper ($T_{FH}$) cells, preventing the formation of functional $T_{FH}$: B cell conjugations, and ultimately leading to a decrease in the production of autoantibodies and the incidence of arthritis[29]. This does demonstrate the potential role of ethanol in reducing the incidence of arthritis. Moreover, the metabolism of ethanol in vivo and the effect of its metabolite acetate on immune cells should be further investigated. Our study showed that consumption of 10% (v/v) ethanol was not toxic to the liver, at

least over the whole animal cycle. After ethanol treatment, the activity of neutrophils was affected by acetate, a metabolite of gut microbiota that plays an important protective role in the pathogenesis of CIA mice.

It has been shown that NETs are associated with a variety of autoimmune diseases, and Ritika Khandpur et al. demonstrated that NETs production was indeed increased in RA patients, promoting the expression of pro-inflammatory genes in fibroblast-like synovial cells[30]. This is consistent with our finding that neutrophils contributed to the pathogenesis of RA by producing NETs. Ethanol treatment reduced the infiltration of various immune cells into the joints and inhibited the formation of NETs in CIA mice. Similar evidence was shown for changes in MPO-DNA in the serum of CIA mice. Gautam Sule found that inhibition of the ERS sensor IRE1 neutralized mitoROS-mediated mitoDNA oxidation in neutrophils, thereby reducing NETs in lupus[19]. Therefore, the protective effect of ethanol on CIA mice may be mediated through reducing ERS in neutrophils and inhibiting the formation of NETs.

Microorganisms and their metabolites can modulate the innate and adaptive immune responses, and dysregulation of the microbiota, such as a higher proportion of lactobacillus, has been observed in both RA patients and experimental arthritic rodents[31]. In this study, a decrease in the abundance of *Lactobacillaceae* was observed after ethanol treatment in CIA mice. In addition, after ethanol treatment, *Muribaculaceae* increased, while *Desulfovibrio*, *Roseburia*, and other conditional pathogenic bacteria decreased. These changes in gut microbiota induced by ethanol treatment were critically associated with the alleviated manifestations of CIA mice. *Muribaculaceae*, as a genus producing SCFAs, produces a large amount of acetate and propionate[32]. Conditional pathogenic bacteria such as *Desulfovibrio* and *Roseobacter* make full use of acetic acid, lactic acid, and fatty acids as carbon sources, reducing the available energy in the body. *Muribaculaceae*, which belongs to the *Bacteroideae*[33], plays an important role in maintaining the integrity of intestinal barrier function and the ability to display anti-inflammatory activity. In this study, although ethanol treatment alone caused the remodeling of intestinal microorganisms, we speculate that it is only when the inflammatory environment and ethanol act at the same time that they can stimulate the abundance of the beneficial bacteria of *Muribaculaceae* to reverse the situation and return the body to a normal state.

The gut microbiota can respond to ethanol intake by activating acetate catabolism rather than by directly metabolizing ethanol. Blood acetate concentrations are elevated during ethanol treatment[34]. In further experiments, we found that ethanol treatment altered the gut microbiota of CIA mice and increased the production of SCFAs such as acetate. Some GPCRs, including GPR41, GPR43, and GPR109A, can be activated by SCFAs. GPR41 is primarily activated by propionate and butyrate, GPR43 binds both

acetate and propionate equally, and GPR109A is a butyrate receptor. In our study, the measurement of SCFA receptors in the gut of mice revealed that ethanol treatment markedly activated GPR43. GPR43 is highly expressed in neutrophils, and SCFAs can act on GPR43 to regulate inflammation[35–37]. Ethanol may play an important role in alleviating CIA mice by regulating the activity of neutrophils through the regulation of acetate, a metabolite of gut microbiota.

Finally, the specific mechanism of acetate was verified in vivo and in vitro. As a multifunctional protein, GRK2 can act as a signal transduction hub by modulating the signal transduction of GPCRs, as well as by phosphorylating or directly interacting with a large number of non-proteins. Many studies have shown that G protein-coupled receptor kinases negatively regulate GPCRs-mediated signal transduction by phosphorylating GPCRs and mediating receptor desensitization[38,39]. In vitro experiments of our study confirmed that acetate activated GPR43 in neutrophils, and reduced ERS in neutrophils to inhibit the formation of NETs by promoting the phosphorylation of GRK2 at S670. In vivo, acetate supplementation alleviated inflammation CIA mice, especially after disrupting the gut microbiota. Acetate profoundly affected various immune cells in the joints of CIA mice, with the most prominent effect on neutrophils. In recent years, the regulation of inflammatory responses by the gut microbiota and the chemoattractant receptor GPR43 has received increasing attention[35]. The role of the gut microbial metabolite SCFAs in suppressing the systemic immune response is gradually being understood, especially in RA[40]. Considering the effect on lymphocytes, butyrate is currently considered to be the most effective SCFA for RA treatment[22,41].Our study shows that acetate produced by gut microbial metabolism of ethanol affected neutrophil activity. Acetate reduces IRE1-dependent ERS-induced NETs through GPR43-GRK2 inhibition, which gives an important and interesting insight in neutrophil biology in RA. Due to the complexity of RA pathogenesis, the production of acetate by the hepatic metabolism of ethanol and the direct or indirect effects of ethanol and acetate on the metabolism, migration, and phagocytosis of neutrophils cannot be ignored. Further studies are needed to better explain the inhibitory effects of acetate on NETs, and unraveling these mechanisms will be of great benefit to further understanding the effects of nutrition on immunity.

Furthermore, current research suggests that even light ethanol consumption can potentially have adverse effects on various diseases, hastening the advancement of conditions such as cardiovascular diseases, liver diseases, and malignant tumors[42,43]. In this study, even relatively low doses of alcohol intake caused weight loss in mice. As a result, it is of utmost importance to carefully assess the advantages of ethanol consumption in light of its detrimental impact on other diseases. As for the adverse effects of EtOH consumption on behaviors of the mice, some basic studies have confirmed that low-dose long-term EtOH intake can increase the occurrence of spontaneous exploratory behaviors (rearing and locomotor activity) through open-field test in rodents[44,45]. Furthermore, previous studies have shown that low dose ethanol treatment can produce reward motivation[46], and have more protective effects, as it reduces inflammation and increases production of neurotrophic factors[47]. We conducted extensive literature review to inform our experiments and established a low-dose ethanol consumption model, but there is still limitation. Therefore, exploring intervention measures that can replicate the beneficial effects of ethanol treatment on rheumatoid arthritis while avoiding its potential side effects becomes a more appealing therapeutic option.

## Conclusions
In summary, the results of this study reveal an important role of ethanol-regulated gut microbial metabolite acetate in RA. Ethanol treatment altered the gut microbiota of mice to produce more acetate. Importantly, acetate affected neutrophil activity, reduced IRE1-dependent ERS, and inhibited NETs formation, thereby playing a protective role in the development of CIA mice (Fig. 8). Our findings may provide insights into a mechanism and therapy for RA.

## Methods
**Human samples**. Human knee synovial tissues used in this study were obtained from patients with RA ($n = 8$) and patients with osteoarthritis (OA) ($n = 5$) undergoing total joint arthroplasty at the department of orthopedics from The Second People's Hospital of Hefei (Hefei, China). The diagnostic criteria for RA patients adopted the 2010 classification criteria of the American College of Rheumatology and the Federation of European Rheumatology Societies. The diagnostic criteria for OA refer to the guidelines for bone and joint diagnosis and treatment updated in 2018 by the Orthopedic Branch of the Chinese Medical Association. Blood samples were taken from RA patients and healthy volunteers. Healthy volunteers had no history of arthritis. During the experiment, all patients gave written informed consent, and the experimental protocol involving human subjects was approved by the Clinical Medical Research Ethics Committee of the First Affiliated Hospital of Anhui Medical University (approval number: 2022275). All ethical regulations relevant to human research participants were followed. The information of RA patients was provided in Table 1.

**Animal**. DBA/1 mice were purchased from Gem Pharmatech (Nanjing, China). 7–8-week-old male DBA/1 mice were used for experiments, which is the week age usually used in most RA modeling experiments[41]. All mice were maintained under specific pathogen-free conditions at 25 °C with 12 h light and dark cycles in accordance with current ethical regulations for animal care and use in China. In the experiment, caging bias was taken into account. Instead of feeding them in an independent ventilated cage, they were all placed in the same environment open to the outside. This study was approved by the Experimental Animal Ethics Committee of Anhui Medical University (approval number: 20220454).

**Ethanol and acetate treatment**. For mice ingested with ethanol, 10% (v/v) ethanol was gradually added to the drinking water of DBA/1 mice before the initial immunization[29,48]. All feedings were changed every 3 days and continued throughout the experiment. For acetate-treated mice, acetate (100 mM)[49] was added to the drinking water of DBA/1 mice starting from day 0 after the initial immunization, and the freshly prepared aqueous solution was replaced every 3 days[22]. Except for drinking water and the variables used to build the model, other treatments were the same for different groups of mice.

**Antibiotic cocktail (ABX) treatment**. Adult mice (8 weeks) gut microbiota were depleted by 3–4 weeks of antibiotic treatment. Add ampicillin (1 g/L), vancomycin (0.5 g/L), neomycin (1 g/L), metronidazole (0.5 g/L) to the formula, and filter with a 0.22 μm filter. Mice were given intragastric administration at a rate of 0.1 mL/10 g at intervals of 24 h each time. According to the weight recovery after ABX treatment, adjust the ABX treatment time. All antibiotics were of United States Pharmacopeia (USP) grade, or minimum cell culture grade. Fecal samples were freshly collected at the end of each experiment, plated on Brucella agar plates with 5% sheep blood, and cultured aerobically and anaerobically to test whether gut microbiota was successfully eliminated after ABX treatment.

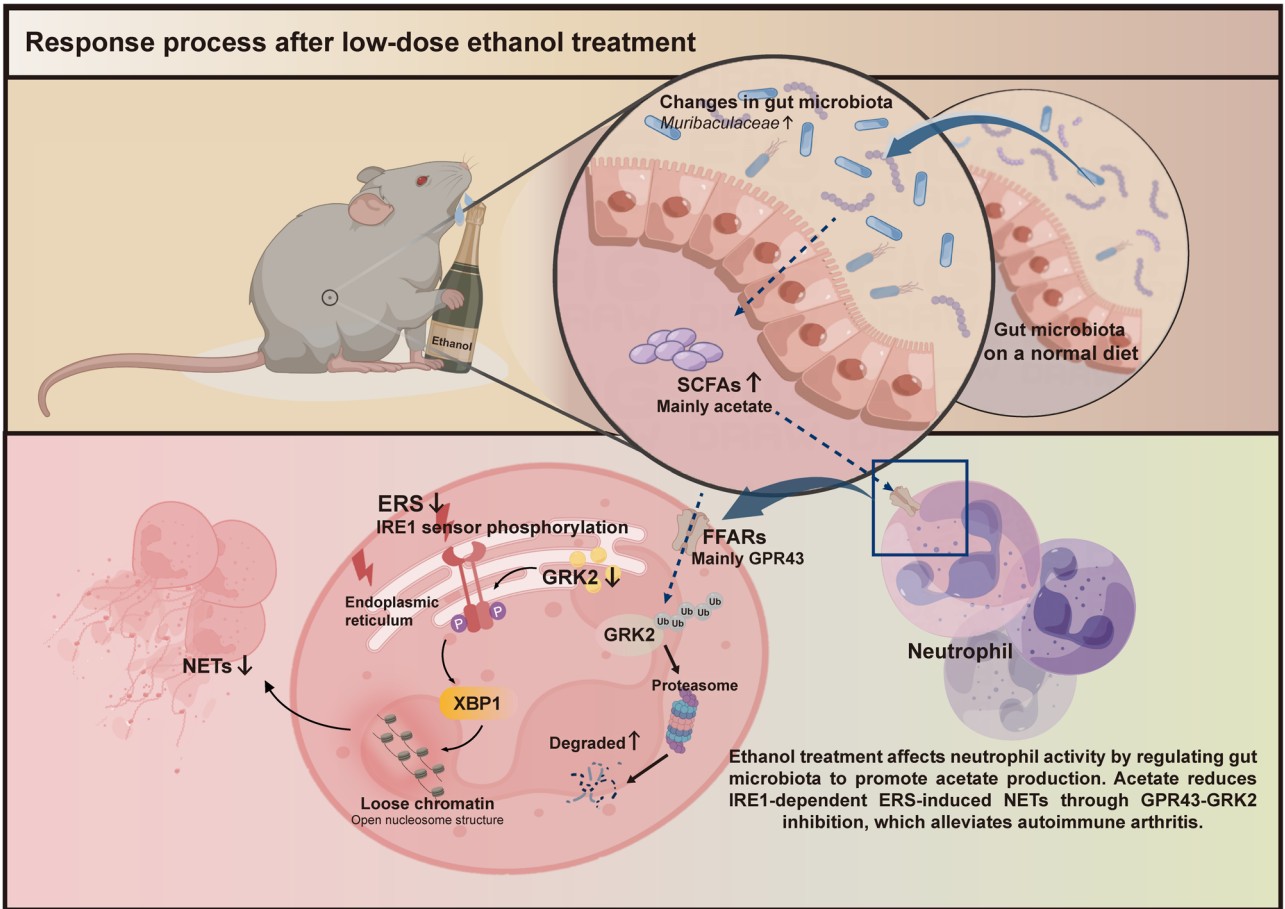

**Fig. 8 Response process after low-dose ethanol treatment.** Low-does ethanol treatment alters the gut microbiota, among which *Muribaculaceae* increases significantly, and the levels of SCFAs especially acetate in serum increases accordingly. Acetate affects the activity of neutrophils by acting on GPR43 on neutrophils, inhibits the formation of NETs by inhibiting the activation of IRE1 phosphorylation and reducing ERS in neutrophils, thereby alleviating the severity of RA.

| Table 1 The information of RA patients. | |
| --- | --- |
| **Group** | **Number (mean ± SD)** |
| Male/Female | 3/5 |
| Age (year) | 59.13 ± 9.10 |
| ESR (mm/h) | 95.75 ± 27.05 |
| CRP (mg/L) | 57.16 ± 31.26 |
| RF (IU/mL) | 180.88 ± 131.61 |
| Anti-CCP | 503.64 ± 627.97 |

**CIA model establishment**. Chicken type II collagen (C9301, Sigma) was fully emulsified with complete Freund's adjuvant containing 5 mg/mL killed mycobacterium tuberculosis (Biological product institute, China), and 100 μL collagen after emulsification was injected subcutaneously into the tail of 8-week-old male DBA/1 mice on day 0. On the 21st day, 100 μL was injected in the same way to enhance immunity. Clinical severity of arthritis was assessed on days 21–49 after the first immunization.

Arthritis index scoring standard: 2 independent blind observers scored the mouse arthritis index every day, scoring from three aspects of overall body score, hind paw swelling score and arthritis score respectively. Overall body scoring criteria: 0 = no paw swelling; 1 = erythema and swelling in one paw; 2 = erythema and swelling in two paws; 3 = erythema and swelling in three paws; 4 = erythema and swelling in all paws. The mice had a maximum overall body score of 4. Hind paw

swelling scoring criteria: 0 = normal paw; 1 = one swelling in the middle of five fingers/toes and wrist/ankle; 2 = two swellings in the middle of five fingers/toes and wrist/ankle; By analogy, the highest score of each paw is 6, and the score of paws swelling number of each mouse is the sum of all paw scores, and the highest hind paw swelling score is 24. Arthritis score: 0 = normal paw; 1 = erythema and mild swelling in the ankle/wrist; 2 = erythema and mild swelling extending from the ankle/wrist to the mid-hind paw/mid-forepaw; 3 = erythema and moderate swelling extending from the digits to the metatarsophalangeal ankles; 4 = erythema and severe swelling of entire paw or ankylosis of the limb. The arthritis score for each mouse was the sum of all paw scores. The highest arthritis score per mouse is 16.

**Cell culture and treatment**. The HL-60 cells (Procell Life Science & Technology Co.,Ltd, China) were cultured in IMDM medium, supplemented with 20% fetal bovine serum and 1% penicillin-streptomycin solution, in a 37 °C, 5% $CO_2$ cell incubator. 1.3% DMSO was added to the culture medium for 7 days to induce HL-60 cells into neutrophil-like differentiation of HL-60 (dHL-60) cells to study the biological function of neutrophils. After incubation with CD15-PE antibody, flow cytometry was used to identify whether the induction was successful. Cells were treated with the following stimulants or reagents: acetate (ACE) (1, 5, 10, 25 mM; 12 h); Phorbol-12-myristate-13-acetate (PMA, NETs

inducer) (20 nM; 4 h); Thapsigargin (TG, ERS agonist) (1 μM; 4 h)[50–52].

**Human neutrophil purification**. Blood from RA patients and healthy volunteers was collected by standard blood collection techniques. Anticoagulated blood was separated by density gradient centrifugation using the Human Peripheral Blood Neutrophil Isolation Kit (P9402, Solarbio). A slightly turbidities liquid layer (enriched neutrophils) was precipitated by cell washing solution, and then red blood cells were lysed by red blood cell lysis solution[53]. After 3 times of washing to remove lysates and debris in red blood cells, precipitated cells were taken as required neutrophils. Neutrophil preparations were at least 95% pure as confirmed by both flow cytometry and nuclear morphology.

**Histological analysis of joint and spleen**. Histopathological features were evaluated using Hematoxylin-eosin (H&E) staining. Isolated hind legs and spleens were fixed in 4% paraformaldehyde for 24 h. The hind legs were subsequently decalcified for one month, while the spleens were directly embedded in paraffin. The degree of inflammation and bone erosion in the knee and ankle joints was assessed by two independent investigators in a blinded manner, with criteria such as synovial hyperplasia, inflammatory cell infiltration, pannus formation, and bone and cartilage damage being considered. In addition, H&E staining of the spleen was also scored blind by two independent investigators on criteria including size of the spleen, number of germinal centers, and leukocyte infiltration. Spleen coefficient is a kind of organ coefficient, calculated by the ratio of spleen weight to body weight of the experimental animal.

**Safranin O staining**. After deparaffinization, sections were immersed in Safranin O staining solution for 3 min, washed with distilled water for 1 min, immersed in fast green staining solution for 2 min, washed with distilled water for 1 min, differentiated with 1% glacial acetic acid for 1 min, dehydrated with 95% ethanol, and discolored with xylene. Finally, the morphological changes in the sections were observed after mounting with neutral gum.

**Immunoblotting and immunoprecipitation**. An appropriate amount of ground cells was added to lysate (10 times volume) and centrifuged at 4 °C, 12000 rpm for 10 min. The supernatant was collected and the protein concentration determined with the BCA method, and equal amounts of lysates were used for immunoblotting and immunoprecipitation. Protein G agarose beads were added into samples and incubated on a rotor at 4 °C. After washing five times with the lysis buffer, the immunoprecipitates were eluted by boiling with the loading buffer containing β-mercaptoethanol for 10 min. Next, SDS-PAGE was performed to separate total protein. The gel was then transferred to PVDF membranes and blocked with skim milk for 2 h. Afterward, the corresponding primary antibodies [GRP78 (ER40402, HUABIO), p-IRE1 (R26310, Zen BioScience), IRE1 (220399, Zen BioScience), CHOP (381679, Zen BioScience), ATF6 (R26445, Zen BioScience), XBP1 (381710, Zen BioScience), H3cit (ab1791, Abcam), MPO (66177-1-lg90, Proteintech), GPR43 (19952-1-AP, Proteintech), GRK2 (R26820, Zen BioScience), p-GRK2 (AF3697, Affinity Biosciences), Ubiquitin (382766, Zen BioScience), β-Actin(AF7018, Affinity Biosciences)] were added, and the mixture was incubated overnight at 4 °C. The membranes were washed with TBST 3 times; the secondary antibody was added, incubated at room temperature for 1 h, washed 3 times with TBST, and developed. Image software was used to analyze the gray value of each band, and the control protein was used to normalize the gray values of the target protein for statistical analysis.

**Immunohistochemical staining**. Mice were euthanized under excessive anesthesia, knee joints were fixed and decalcified, dehydrated with graded alcohol, paraffin embedding, and sliced 5 μm. The tissue sections were placed in an oven at 55 °C overnight, immersed in xylene and dewaxed with gradient ethanol, washed 3 times with PBS, penetrated for 15 min, washed with PBS for 3 times, repaired with antigen for 15 min, washed with PBS for 3 times, sealed with endogenous peroxidase for 10 min, rinsed with PBS for 3 times, and sealed with serum for 15 min. Then, the primary antibody [MPO (R25062, Zen BioScience)] was added and the mixture was incubated overnight at 4 °C. The next day, after incubation with biotin-labeled secondary antibody for 20 min at room temperature, the sections were rinsed with PBS 3 times, horseradish enzyme marker was added dropwise to streptavidin working solution, rinsed with PBS 3 times, developed color with DBA, stained with hematoxylin, dehydrated by gradient alcohol and mounted with neutral gum. Finally, the results of tissue staining were observed under digital slice scanning system Pannoramic MIDI (3D HISTECH, Hungary).

**Immunofluorescence microscopy**. The dHL-60 cells were adhered to the polylysine-coated 1.5 mm overlay of a 24-well plate. As previously shown, after pretreatment with acetate (10 mM) for 12 h, dHL-60 cells were stimulated with PMA (20 nM). After 4 h, they were fixed with 3.7% paraformaldehyde at 4 °C overnight, and then permeabilized with PBS + 0.1% Triton X-100 for 15 min. Use primary antibody [MPO (66177-1-lg 90, Proteintech) and/or p-IRE1 (Ser724) (R26310, Zen BioScience)] in staining buffer (PBS + 0.1% Triton X-100, 5% BSA and 10% normal goat serum) for primary staining. After the antigen retrieval of the tissue sections was completed, they were incubated with 5% normal goat serum in PBS for 1 h at room temperature to block the binding of non-specific antibodies. Slides were stained overnight at 4 °C with the following primary antibodies: MPO (66177-1-lg 90, Proteintech) and/or p-IRE1 (Ser724) (R24754, Zen BioScience) and/or H3cit (ab219407, Abcam). Secondary antibodies conjugated to Alexa Fluor 594 (711-585-152, Jackson ImmunoResearch) and Alexa Fluor 488 (715-545-151, Jackson ImmunoResearch) were used according to the manufacturer's instructions. The second antibody was incubated for 1 h in light and washed 3 times by PBS. Then sections were incubated DAPI for 10 min, wash 3 times, and dry water stains. Finally, anti-fluorescence quench agent was added to seal the slide and digital slice scanning system Pannoramic MIDI (3D HISTECH, Hungary) was used for scanning imaging.

**Scanning electron microscopy**. The dHL-60 cells were adhered to the polylysine-coated 1.5 mm overlay of a 24-well plate. As previously shown, after pretreatment with acetate (10 mM) for 12 h, dHL-60 cells were stimulated with PMA (20 nM). After 4 h, fixed in 2% glutaraldehyde (which was prepared in phosphate buffered saline) for 2 h. The sample then dehydrated through a series of ethanol gradients, starting with 20% and ending with 100% (20, 40, 60, 80 and 100%). The sample were submerged in each percentage of alcohol for approximately 30 min. Then, imbibed in 100% acetone before being critical point dried, gold coated on an aluminum stub, and viewed under a GeminiSEM300 (ZEISS, Germany) scanning electron microscope.

**Live cell confocal microscopy**. After the pre-treated cells were replaced with serum-free media, $2 \times 10^5$ dHL-60 cells per well were inoculated into confocal glass petri dishes (Biosharp, BS-15-

GJM), and Sytox-green fluorescent dye was added. The dishes were mounted on a heated stage within a temperature-controlled chamber maintained at 37 °C and constant $CO_2$ concentrations (5%) and infused using a gas incubation system with active gas mixer. Optical sections were acquired through the center of the cells by sequential scans of excitation 488 nm or brightfield/differential interference contrast on a TCS SP5 confocal microscope (Leica, Germany) using a 40× air objective and Leica Application Suite X software (Leica, Germany). Images were acquired at 3 mins/frame for 4 h.

**Flow cytometry and cell sorting**. The mouse feet were cut into tissue blocks of 3–4 mm in a 1.5 mL EP tube, and added Hank's equilibrium salt solution (1 mL), collagenase type II (1 g/mL, 2 μL) and $CaCl_2$ solution (3 mM, 5 μL). After incubated at 37 °C for 4 h, taken the supernatant by centrifugation, washed with PBS 3 times after passing through a nylon sieve to obtain dispersed cells. Fluorescent antibodies were added to the treated single cell suspension ($1 \times 10^6$/100 μL), and incubated at 4 °C in the dark for 30 min. Then, after centrifugation at 300 g, PBS was added and the cells were repeatedly washed 3 times. Finally, 300 μL PBS was added to resuspend, and then the results of cell sorting were observed. Isolated cells were stained with the following antibodies: CD45-FITC (553080, BD Pharmingen), CD19-PE (553786, BD Pharmingen), CD3-PE (553063, BD Pharmingen), CD4-FITC (553650, BD Pharmingen), CD11b-FITC (101206, BioLegend), Ly6G-PE (551461, BD Pharmingen), Ly6C-APC (553129, BD Pharmingen).

**DNA extraction and bacterial 16S rRNA sequencing**. Aseptically collected stool samples were stored at −80 °C until use. The E.Z.N.A. Stool DNA kit (D4015, Omega) was employed to extract DNA from the samples following the manufacturer's instructions[54]. The α-diversity and β-diversity of the samples were assessed using various metrics including Chao1, Shannon, Simpson, principal component analysis (PCA), and principal coordinate analysis (pCoA). All indicators in the sample adopt the QIIME 2 analysis process, and call DADA 2 to denoise the Raw Data, which is equivalent to clustering DNA sequences with 100% similarity, only removing and correcting low-quality sequences, algorithm identification and chimerism etc.; the denoised sequence is directly de-redundant to obtain figure (features, a general term foroutU, ASV, etc.) information.

**GC-MS analysis of seven SCFAs**. The procedure follows the previous description[55]. To prepare the sample, serum was thawed slowly at 4 °C and an appropriate amount of sample and 50 μL 15% phosphoric acid were added to a 2 mL EP tube. Subsequently, 100 μL of internal standard solution (isohexanoic acid, 125 μg/mL) and 400 μL of ether were added to the tube. The sample mixture was homogenized for 1 min and then centrifuged at 12,000 rpm for 10 min at 4 °C. The supernatant was carefully collected and analyzed by machine. The GC analysis was performed on trace 1300 gas chromatograph (Thermo Fisher Scientific, USA). The GC was fitted with a capillary column Agilent HP-INNOWAX (30 m × 0.25 mm ID × 0.25 μm) and helium was used as the carrier gas at 1 mL/min. Injection was made in split mode at 10:1 with an injection volume of 1 μL and an injector temperature of 250 °C. The temperature of the ion source and MS transfer line were 300 °C and 250 °C, respectively. The column temperature was programmed to increase from an initial temperature of 90 °C, followed by an increase to 120 °C at 10 °C/min, and to 150 °C at 5 °C/min, and finally to 250 °C at 25 °C/min which was maintained for 2 min. Mass spectrometric detection of metabolites was performed on ISQ 7000 (Thermo Fisher

Scientific, USA) with electron impact ionization mode. Single ion monitoring (SIM) mode was used with the electron energy of 70 eV. Peak determination and peak area integration were analyzed by MassHunter Worksta software Version B.8.0 (Agilent, USA).

**ELISA**. The samples were diluted and assayed according to the Mouse MPO-DNA ELISA kit (YJ303625, MLBIO) operating instructions. Samples, standard products, and HRP labeled detection antibodies were added to the coated micropores that were pre-coated with captured antibodies, then incubated and thoroughly washed. Color was produced with the substrate TMB, which was converted to blue by the catalysis of peroxidase and to the final yellow by the action of acid. The shade of the color was positively correlated with the sample. The absorbance (OD value) was measured at 450 nm wavelength and the sample concentration was calculated.

**Real-time quantitative PCR (RT-qPCR)**. TRIzol Reagent (15596026, Thermo Fisher Scientific) was used to extract total RNA from cells or tissues to determine RNA concentration and purity. Samples of qualified purity were used to perform reverse transcription and PCR amplification tests in accordance with the manufacturer's instructions. The PCR reaction volume was 20 μL, and the amplification conditions were predenaturation 95 °C, 5 min; cycle (40 times) 95 °C, 10 s; 60 °C, 35 s; and melting curve 95 °C, 15 s; 60 °C, 60 s; and 95 °C, 15 s. The *β-Actin* gene was used as an internal reference (Table 2), and the relative quantitation of gene expression was analyzed using the $2^{-\Delta\Delta CT}$ method.

**Statistics and reproducibility**. PRISM V.9.0.0 (GraphPad, USA) was used for statistical analysis. Data in figures were shown as the mean ± SEM. Statistical comparisons were performed using Student's *t* test for comparison between two groups or for paired comparisons, one-way ANOVA followed by Tukey post hoc test when more than two groups under same condition were involved, and two-way ANOVA followed by Sidak's or Tukey's post hoc test for comparison between two or more groups under two

**Table 2 Primer sequences used in this study.**

| Gene | F/R | Sequence |
|---|---|---|
| *Il-1β* | F | 5′-TGCCACCTTTTGACAGTGATG-3′ |
| | R | 5′-TGATGTGCTGCTGCGAGATT-3′ |
| *Il-17a* | F | 5′-CTGGACTCTCCACCGCAATG-3′ |
| | R | 5′-GGACCAGGATCTCTTGCTGG-3′ |
| *Ifn-γ* | F | 5′-GGAGGAACTGGCAAAAGGATG-3′ |
| | R | 5′-GTTGCTGATGGCCTGATTGT-3′ |
| *Tnf-α* | F | 5′-GATCGGTCCCCAAAGGGATG-3′ |
| | R | 5′-CCACTTGGTGGTTTGTGAGTG-3′ |
| *Ly6g* | F | 5′-GGGAGGGGCTGAGAGAAAGTA-3′ |
| | R | 5′-AGGGCTGCACAGATAAAACTTCC-3′ |
| *Cxcl1* | F | 5′-CCCAAACCGAAGTCATAGCCA-3′ |
| | R | 5′-TTCTTAACTATGGGGGATGCAG-3′ |
| *Cxcl2* | F | 5′-TTGTCTCAACCCCGCATCG-3′ |
| | R | 5′-GGTCAGTTGGATTTGCCATTTT-3′ |
| *Ccl2* | F | 5′-TCAAACTGAAGCTCGCACTCT-3′ |
| | R | 5′-GGCATTGATTGCATCTGGC-3′ |
| *Gpr41* | F | 5′-GTGACCATGGGGACAAGCTTC-3′ |
| | R | 5′-CCCTGGCTGTAGGTTGCATT-3′ |
| *Gpr43* | F | 5′-GGCTTCTACAGCAGCATCTA-3′ |
| | R | 5′-AAGCACACCAGGAAATTAAG-3′ |
| *Gpr109a* | F | 5′-GGCGTGGTGCAGTGAGCAGT-3′ |
| | R | 5′-GGCCCACGGACAGGCTAGGT-3′ |
| *β-Actin* | F | 5′-CATTGCTGACAGGATGCAGAAGG-3′ |
| | R | 5′-TGCTGGAAGGTGGACAGTGAGG-3′ |

conditions. *p*-value less than 0.05 was considered significant. For RT-qPCR analyses, mRNA levels were normalized to the mean of the control condition. For Western blot analysis and cell culture experiments, values at different time points or different conditions were normalized to the control condition on the same blot, or the same cell culture experiment. For all statistical analyses of normalized data, the log (on the basis of 2) of each data point was calculated and statistical analyses were done with these transformed values.

**Inclusion and ethics**. All experimental procedure were conducted in accordance with ethical regulation for animal care and use in China and approved by the Animal Ethical Council of Anhui Medical University and the Clinical Medical Research Ethics Committee of the First Affiliated Hospital of Anhui Medical University. Animal welfare and experimental procedures were strictly in accordance with the guidelines for the care and use of laboratory animals.

**Reporting summary**. Further information on research design is available in the Nature Portfolio Reporting Summary linked to this article.

## Data availability

All data that have been generated or analyzed during this study are included in the relevant databases and the associated supplementary files. All uncropped and unedited Western Blots have been merged into Supplementary Fig. 5. The Fig. 8 was created with BioRender.com (agreement number: LR25YSBQS3). The NCBI SRA accession number for the 16S rRNA sequencing data in this paper is PRJNA1026760. Supplementary videos (https://doi.org/10.6084/m9.figshare.22710202) and the source data for the graphs (https://doi.org/10.6084/m9.figshare.24286627) are available.

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

## Acknowledgements

We acknowledge the support received from the National Natural Science Foundation of China (No. 82204403; No. 82373878); China Postdoctoral Science Foundation (No. 2022M72199); Anhui Natural Science Foundation (No. 2308085MH312); Educational Commission of Anhui Province of China (No. 2023AH040080); Key Research and Development Plan of Anhui Province in 2021-Population Health Project (No. 202104j07020032) and open project of Key Laboratory of Anti-inflammatory Immune Drugs of Ministry of Education (No. KFJJ-2021-07). Especially, L.J. wishes to appreciate the invaluable support and encourage from Chuan Huang throughout the completion of this article, you light up my life.

## Author contributions

L.J., Z.Z., Y.Z., M.Z., L.L., and P.P. conducted most experiments, analyzed data, and wrote manuscripts. Y.Z., H.W., L.X., H.Z., and Y.Y. conducted some experiments. D.M. and J.H. helped to the bioinformatics analyses. L.Z. and X.Z. provided knowledge for the design and conduct of the experiment, data processing, and manuscript writing.

## Competing interests

The authors declare no competing interests.
