## [Peer Review File · Communications Biology]

Reviewers' comments:

Reviewer #1 (Remarks to the Author):

The group of Lingling Zhang reports in a comprehensive study how low-dose EtOH consumption is beneficial in arthritic mice. The group reports that the ethanol-derived metabolite acetate is responsible for activation of the GPCR GPR43, which efficiently inhibits ER-stress and NET formation in neutrophils, which is known to contribute to disease progression. Although some findings in this study are not novel, such as acetate is derived from EtOH and activates GPR43, NET formation is inhibited by acetate, NET formation contributes to RA and ethanol consumption is beneficial in RA, this story shows in a detailed way the mechanisms of EtOH consumption-caused amelioration of CIA in mice and also includes supporting experiments on human neutrophils.

The main novelty, to my knowledge, is the mechanistic detail of acetate-induced inhibition of NET formation in neutrophils through GPR43-GRK2 and inhibition of IRE1a-dependent ER-stress, which gives an important and interesting insight in neutrophil biology in RA.

The work is convincing, but it should be clearly indicated in the text what is known and what is new.

(1) The flow cytometry data shown are not clear, the shown plots are not matching the data shown in the graphs next to it, gating is not clear. The gates are not gating on the population indicated, e.g. Cd11b+Ly6C+ is gating on all Cd11b+, not just Ly6C+, same for Ly6G and CD4/CD3,... in figure 7. Labeling is additionally too small. Same in figure 7.

(2) Line 308: The authors are stating: "Ethanol treatment was confirmed by measuring mouse body weight and dietary intake every three days from day 1 (Supplementary Fig. 1A and 1B)"  how can EtOH feeding be confirmed by body weight and water intake measurement? Shouldn't it be confirmed by blood alcohol measurement? It would be of interest how much alcohol was actually consumed daily by the mice. It is not discussed why the drinking water uptake and food intake in the EtOH group is significantly lower compared to NC, is the significant lower intake of food and water concerning? It should be at least mentioned and discussed and not just ignored. Does the EtOH consumption has any adverse effects on behavior of the mice? Some field tests, like orientation and activity assays could be included.

(3) How was the spleen coefficient calculated? Include it in methods and a short description in the text would be helpful.

(4) Figure legends should be more detailed to fully understand the shown data. Typos in axis labeling need to be corrected.

(5) If there is no significance between groups, especially in 16S rRNA sequencing data, it should not be stated as significant difference in the text. see Shannon index, figure 4a, no significant difference, compare to line 395

(6) Picture of paws should be shown as larger images, especially in figure 7.

(7) line 349: The last sentence is not complete.

(8) It would be helpful to include the used methods in the text, e.g., line 372: activation of IRE1a  how is it measured

(9) line 478 ff: expression of GPR43 was increased in the presence of acetate, indicating that GPR43 can be activated by acetate  just because expression is increased, doesn't mean the receptor is activated by acetate, this statement needs to be changed.

(10) line 540: What are TFH cells? Introduce new terms in the text.

(11) The measured concentration of acetate in serum of CIA-EtOH mice is around 20 uM, but the inhibitory effect of acetate seen on neutrophils in vitro is at 5-10 mM. Is the acetate concentration in the joints expected to be higher or how can it be explained that only such a high concentration of acetate has inhibitory effects on NETs. Can an acetate serum concentration in the millimolar range be

reached under physiological conditions?

Reviewer #2 (Remarks to the Author):

This study investigates the association between alcohol (ethanol) consumption and rheumatoid arthritis (RA), as previous studies have shown a negative correlation. Although this is an interesting concept, the work reported is largely correlative rather than causative and has some gaps in logic and/or rationale. The authors suggest ethanol intake modulates the microbiome, increasing acetate producing bacteria, therefore decreasing NETosis via the effect of acetate on ER stress. They do not address the production of acetate directly from ethanol by cells, which may provide a more likely source for the increased acetate levels in mice treated with ethanol. This should be addressed. A major flaw of the work presented lies in the quality of the figures, which unfortunately meant I could not accurately review the manuscript in its entirety, as all of figure 7 was too blurred to interpret.

Specific points below:

Methods:

- 1) Please comment on why male mice were used, when females would more accurately recapitulate RA which has a female bias
- 2) please comment on the age of mice used
- 3) please clarify how the mice were caged - groups should be mixed to avoid caging bias
- 4) please clarify where the NC and EtOH control groups were generated. Were these mice housed with CIA groups, did they undergo similar procedures etc. This point is particularly relevant to the microbiome assessment, as the CIA treatment, handling during disease scoring etc would have increased exposure to bacteria
- 5) is the concentration of acetate used 100mM physiologically relevant to the concentrations observed in the study?
- 6) Figure 2A (flow cytometry) needs justification of gating strategy (e.g FMO or unstained controls) as the gates are not convincing. Also the gates shown do NOT show what is denoted in the legend and text. I.e. Gating shown is not Ly6c+ OR Ly6G+ but CD11b+ only. In addition CD11b+Ly6c+ cells would include monocytes and macrophages, not macrophages as stated in the text

General

- 7) line 349 is missing
- 8) Figure 1C-E is very poor quality
- 9) Figure 7 is so blurry I can't interpret any of it
- 10) the flow cytometry suggests changes in monocytes and T-cells as well as neutrophils, why were these cells not investigated as associated with the ethanol phenotype?
- 11) Fig 1K the effect on inflammatory cytokines more significant than Ly6G – suggests this may be the cause for reduced severity of disease rather than via NETs
- 12) Line 414 Muribaculaceae was not increased with ethanol only compared to CIA only. Please clarify and justify why ethanol without CIA does not affect Muribaculaceae. 566 repeated
- 13) Figure 1: What is total body score? This is not defined, but used to determine incidence. It is unusual to define incidence based on body score and not swelling - swelling appears to have a similar time of onset in figure D.
- 14) Fig 2D is cropped
- 15) Fig 6: EtOH increases Netosis, please clarify as this does contradict the overall claims.

Point-by-point responses to Reviewers' comments

Reviewer #1

We thank the reviewer for remarking that our work is “comprehensive” and “convincing”. We appreciate the statement that our study “gives an important and interesting insight in neutrophil biology in RA” and hope our responses to the queries below will be satisfactory.

Remark #1:

The group of Lingling Zhang reports in a comprehensive study how low-dose EtOH consumption is beneficial in arthritic mice. The group reports that the ethanol-derived metabolite acetate is responsible for activation of the GPCR GPR43, which efficiently inhibits ER-stress and NET formation in neutrophils, which is known to contribute to disease progression. Although some findings in this study are not novel, such as acetate is derived from EtOH and activates GPR43, NET formation is inhibited by acetate, NET formation contributes to RA and ethanol consumption is beneficial in RA, this story shows in a detailed way the mechanisms of EtOH consumption-caused amelioration of CIA in mice and also includes supporting experiments on human neutrophils.

The main novelty, to my knowledge, is the mechanistic detail of acetate-induced inhibition of NET formation in neutrophils through GPR43-GRK2 and inhibition of IRE1 α -dependent ER-stress, which gives an important and interesting insight in neutrophil biology in RA.

The work is convincing, but it should be clearly indicated in the text what is known and what is new.

Reply #1:

Thank you for your valuable comments. As suggested, we have extensively revised the

introduction and discussion sections to indicated what is known and what is new in this study, including ethanol consumption is beneficial in RA (Page 2, Line 44-49), as follow:

Ethanol (C₂H₅OH, EtOH; commonly referred to as alcohol) as part of the dietary composition has been shown to have effects on innate and adaptive immunity, with prolonged high-dose alcohol consumption impairing host defenses (Szabo et al., 2015). In fact, moderate alcohol consumption has long been recognized as a protective factor in the pathogenesis of autoimmune diseases such as RA (Lu et al., 2014; Kallberg et al., 2009), but the specific mechanism of action has been unclear.

The statement that acetate is derived from EtOH and activates GPR43 has been add (Page 2, Line 55-60) as follow:

It has been reported that low-dose ethanol could be metabolized by the gut microbiota into SCFAs, in which the ratio of acetic acid, propionic acid, and butyric acid was about 60:20:20 (Sivaprakasam et al., 2013). SCFAs can bind to receptors on innate immune cells, such as free fatty acid receptor 3 (GPR41), free fatty acid receptor 2 (GPR43) and GPR109A highly expressed on the surface of innate immune cells represented by neutrophils (Carretta et al., 2021).

The statement that NETs formation contributes to RA has been added (Page 3, Line 79-81) as follow:

Although NETs are important host defense mechanisms (Tillack et al., 2012), their abnormal formation and clearance can lead to tissue damage and abnormal activation of the immune system in autoimmune diseases such as RA (Papayannopoulos et al., 2018).

The innovation of this research has been added in the discussion (Page 10-11, Line 408-411) as follow:

Our study shows that acetate produced by gut microbial metabolism of ethanol significantly affected neutrophil activity. Acetate reduces IRE1-dependent ERS-

induced NETs through GPR43-GRK2 inhibition, which gives an important and interesting insight in neutrophil biology in RA.

The brief summary of the major results and conclusions has been added (Page 3, Line 93-103) as follow:

In this study, we applied flow cytometry with ultrasonography and X-ray imaging to elucidate the ethanol treatment inhibits the formation of NETs in the joints of RA mice. Then, microbiomic and metabolomic analyses revealed that the ethanol treatment can increase acetate by affecting gut microbes to inhibit RA through gut-joint axis. Meanwhile, acetate reduced ERS in neutrophils and inhibited the formation of NETs by scanning electron microscopy and Western blot. Furthermore, exogenous acetate reversed CIA in mice with exacerbated gut microbial disruption, further confirming that the effect of gut microbial metabolite acetate on neutrophils in vivo is crucial for the immune regulation. This work emphasizes the pivotal role of acetate during RA intervention, which may offer alternative avenues to replicate or induce the joint-protective benefits of ethanol without associated detrimental effects.

Remark #2:

The flow cytometry data shown are not clear, the shown plots are not matching the data shown in the graphs next to it, gating is not clear. The gates are not gating on the population indicated, e.g., Cd11b+Ly6C+ is gating on all Cd11b+, not just Ly6C+, same for Ly6G and CD4/CD3, ... in figure 7. Labeling is additionally too small. Same in figure 7.

Reply #2:

Thank the reviewer for these thoughtful suggestions. We have clarified the flow cytometry data and re-adjusted the gating strategy in figure 2 and 7:

In Figure 2, the flow cytogram of CD11b+Ly6C+ and CD11b+Ly6G+ has been

analyzed using the cross-gate quadrant gate method to analyze the percentage of double-positive cells, and all labels have been enlarged to demonstrate clearly.

In Figure 7, in order to improve the quality of the picture and display the results more clearly, we have re-formatted and put the incidence of CIA mice (Supplementary Fig. 3A), arthritis score (Supplementary Fig. 3B), X-ray images of paws (Supplementary Fig. 3C) and flow cytometry scatter plot (Supplementary Fig. 3D) in Supplementary Figure 3. As for the flow cytogram, we have specifically shown the gating strategy of each gate in the supplementary figure, and indicated the percentage of positive cells of each gate in the figure labeling.

Remark #3:

Line 308: The authors are stating: “Ethanol treatment was confirmed by measuring mouse body weight and dietary intake every three days from day 1 (Supplementary Fig. 1A and 1B)”  how can EtOH feeding be confirmed by body weight and water intake measurement? Shouldn't it be confirmed by blood alcohol measurement? It would be of interest how much alcohol was actually consumed daily by the mice. It is not discussed why the drinking water uptake and food intake in the EtOH group is significantly lower compared to NC, is the significant lower intake of food and water concerning? It should be at least mentioned and discussed and not just ignored. Does the EtOH consumption has any adverse effects on behavior of the mice? Some field tests, like orientation and activity assays could be included.

Reply #3:

Thanks for these detailed comments. Ethanol feeding is confirmed by their daily intake of liquid, which is a mixture of 10% ethanol and 90% water (Azizov et al., 2020; Jonsson et al., 2007). A previous study measured ethanol levels in the mice's serum and found that the consumption of 10% ethanol in drinking water increased the alcohol content in the mice's blood to 0.03 mg/mL (Azizov et al., 2020). Measuring blood

alcohol levels indeed provides a more accurate determination of mice's ethanol intake, however, repeatedly taking blood is challenging to implement in our experimental modeling process and could potentially affect the accuracy of subsequent experimental measurements due to the harm caused to the mice during the process.

Regarding the significantly lower intake of food and water in the EtOH group compared to the NC group, we have added a discussion on the impact of ethanol intake and consumption in the final section (Page 11, Line 418-423, 429-434) as follow:

Furthermore, current research suggests that even light ethanol consumption can potentially have adverse effects on various diseases, hastening the advancement of conditions such as cardiovascular diseases, liver diseases, and malignant tumors (GBD 2020 Alcohol Collaborators; GBD 2016 Alcohol Collaborators;). In this study, even relatively low doses of alcohol intake caused significant weight loss in mice. As a result, it is of utmost importance to carefully assess the advantages of ethanol consumption in light of its detrimental impact on other diseases.

We conducted extensive literature review to inform our experiments and established a low-dose ethanol consumption model, but there is still limitation. Therefore, exploring intervention measures that can replicate the beneficial effects of ethanol treatment on rheumatoid arthritis while avoiding its potential side effects becomes a more appealing therapeutic option.

As for the adverse effects of EtOH consumption on behaviors of the mice, some basic studies have confirmed that low-dose long-term EtOH intake can increase the occurrence of spontaneous exploratory behaviors (rearing and locomotor activity) through open-field test in rodents (Waller et al., 1986; Kimoto et al., 2017). Furthermore, previous studies have shown that low dose ethanol treatment can produce reward motivation (Bryant et al., 2022), and have more protective effects, as it reduces inflammation and increases production of neurotrophic factors (Tizabi et al., 2018). As reviewer suggested, these effects of low dose ethanol have been reported in discussion

(Page 11, Line 423-429).

Remark #4:

How was the spleen coefficient calculated? Include it in methods and a short description in the text would be helpful.

Reply #4:

Thank reviewer for this helpful reminder. As suggested, we have added the description of the method in materials and methods (Page 14, Line 539-541) as follow:

Spleen coefficient is a kind of organ coefficient, calculated by the ratio of spleen weight to body weight of the experimental animal.

Remark #5:

Figure legends should be more detailed to fully understand the shown data. Typos in axis labeling need to be corrected.

Reply #5:

Thank reviewer for this kindly suggestion. We have carefully checked all figures and supplemented the figure legend in detail and corrected typos in axis labeling.

Remark #6:

If there is no significance between groups, especially in 16S rRNA sequencing data, it should not be stated as significant difference in the text. see Shannon index, figure 4a, no significant difference, compare to line 395

Reply #6:

Thank reviewer for pointing out this mistake and we have corrected it (Page 5, Line 198-200) as follow:

As shown in Fig. 4A, there were no significant difference among the diversities of gut microbiota in mice of the control group, CIA group, EtOH group, and EtOH+CIA group.

Remark #7:

Picture of paws should be shown as larger images, especially in figure 7.

Reply #7:

Thank reviewer for this helpful comment. As suggested, we have resized the graphics and the resolution, and carefully adjusted the figure layout to ensure that all components of the graphics were clearly visible.

Remark #8:

line 349: The last sentence is not complete.

Reply #8:

Thank you for your careful comments, and we have completed this sentence (Page 4, line 151-153) as follow:

These data suggest that low-dose ethanol consumption alleviates the clinical manifestations of CIA mice and protects bone from erosion.

Remark #9:

It would be helpful to include the used methods in the text, e.g., line 372: activation of IRE1a  how is it measured

Reply #9:

Thank reviewer for the helpful suggestion. In physiological condition, IRE1 binds to GRP78 on the endoplasmic reticulum. When endoplasmic reticulum stress occurs, IRE1 is activated through autophosphorylation, showing endonuclease activity, cutting the downstream molecule (XBP1) base of IRFE1 signaling pathway, and generating an active spliceosome XBP1s to regulate the downstream. Therefore, the activation of IER1 can be detected by the expression of p-IRE production (Sule, G. et al., 2021; Chen et al., 2022). Base on that, we have supplemented and modified this part (Page 5, Line 174-181) as follow:

When ERS occurs, IRE1 is activated through autophosphorylation, showing endonuclease activity, cutting the downstream molecule (XBP1) base of IRFE1 signaling pathway, and generating an active spliceosome XBP1s to regulate the downstream. However, the role of ERS in the pathogenesis of autoimmune arthritis remains unclear. Our results found that ethanol treatment reduced the phosphorylation of IRE1 (p-IRE1) in neutrophils while reducing the formation of NETs (Fig. 3A). Therefore, activation of IRE1 phosphorylation after involvement in ERS may be involved in autoimmune arthritis by promoting the formation of NETs.

Remark #10:

line 478: expression of GPR43 was increased in the presence of acetate, indicating that GPR43 can be activated by acetate  just because expression is increased, doesn't mean the receptor is activated by acetate, this statement needs to be changed.

Reply #10:

Thank reviewer for this instructive comment, and we have rephrased this sentence (Page 7, Line 278-279) as follow:

And the expression of GPR43 was significantly increased in the presence of acetate.

Remark #11:

line 540: What are TFH cells? Introduce new terms in the text.

Reply #11:

Thank reviewer for this kind reminding. As suggested, the full name has been added before the acronym in the text (Page 9, Line 339-340): T follicular helper (T_{FH}) cells.

Remark #12:

The measured concentration of acetate in serum of CIA-EtOH mice is around 20 uM, but the inhibitory effect of acetate seen on neutrophils in vitro is at 5-10 mM. Is the acetate concentration in the joints expected to be higher or how can it be explained that only such a high concentration of acetate has inhibitory effects on NETs. Can an acetate serum concentration in the millimolar range be reached under physiological conditions?

Reply #12:

The reviewer has raised a meticulous question about the effective concentration of acetate used in the in vitro and in vivo experiments.

In vitro, the stimulator Phorbol-12-myristate-13-acetate (PMA), an effective stimulator, was used to induce the production of NETs, which results in much higher production of NETs than in inflammatory conditions. However, in vivo, NETs are the result of interaction of multiple antigens.

In this part of the study, the specific mechanism of acetate inhibiting NETs was mainly explored through in vitro experiments. Low doses of acetate had no obvious inhibitory

effect on NETs in vitro. Moreover, the duration of action of acetate also varied in in vitro and in vivo experiments. The effect of acetate on neutrophils lasted only 12h in vitro, but the acetate produced by the ingestion of ethanol stay in vivo is a long-acting process. Therefore, the concentration of acetate was increased to 5-10 mM for further in vitro mechanistic studies.

Reviewer #2:

We thank reviewer for commenting the concept of our work is “interesting”, and hope that the following questions can be answered satisfactorily.

Remark #1:

This study investigates the association between alcohol (ethanol) consumption and rheumatoid arthritis (RA), as previous studies have shown a negative correlation. Although this is an interesting concept, the work reported is largely correlative rather than causative and has some gaps in logic and/or rationale.

The authors suggest ethanol intake modulates the microbiome, increasing acetate producing bacteria, therefore decreasing NETosis via the effect of acetate on ER stress. They do not address the production of acetate directly from ethanol by cells, which may provide a more likely source for the increased acetate levels in mice treated with ethanol. This should be addressed. A major flaw of the work presented lies in the quality of the figures, which unfortunately meant I could not accurately review the manuscript in its entirety, as all of figure 7 was too blurred to interpret.

Reply #1:

We thank the reviewer for this important and insightful comment. As the reviewer commented, alcohol have many metabolic pathways to produce acetate in physiological condition. In order to exclude the influence of other metabolic pathways, we established a mouse model of ABX, which disrupted the gut microbiota of CIA mice by using antibiotics and significantly worsened disease symptoms. Mice in the CIA+ABX+ACE group supplemented with acetate showed significantly lower disease symptoms than those in the CIA+ABX group, confirming that gut bacteria are the main source of acetate production. These results are shown in Fig 7, The reviewers may have overlooked this part of results due to the ambiguity of Fig 7 in the original manuscript. Now we have resized the graphics and the resolution, and carefully adjusted all graphic

layout, especially in Figure 7. To ensure clear visibility of all components of Figure 7, the incidence of CIA mice (Supplementary Fig. 3A), arthritis score (Supplementary Fig. 3B), X-ray images of paws (Supplementary Fig. 3C) and flow cytometry scatter plot (Supplementary Fig. 3D) have been adjusted for supplementary figure, and all figures were uploaded as original images for readers to download and review.

Remark #2:

Please comment on why male mice were used, when females would more accurately recapitulate RA which has a female bias.

Reply #2:

Thank reviewer for this insightful suggestion. Females are more likely to get rheumatoid arthritis, but it is reported that no gender difference in CIA mice (Brand et al., 2007). In order to avoid the impact of various hormone fluctuations, such as estrogen, in the female menstrual cycle on the pathogenesis of RA, only male mice are used as experimental subjects (Miyoshi et al.,2018).

Remark #3:

please comment on the age of mice used

Reply #3:

Thank you for your careful comments. The age of mice is between 7 to 8 weeks, and this comment has been added in the text (Page 12, Line 458-460) as follow:
7–8-week-old male DBA/1 mice were used for experiments, which is the week age usually used in most RA modeling experiments (Miyoshi et al.,2018).

Remark #4:

please clarify how the mice were caged - groups should be mixed to avoid caging bias

Reply #4:

Thank reviewer for this thoughtful comment. As suggested, we have supplemented this part of clarification in the materials (Page 12, Line 460-464) as follow:

All mice were maintained under specific pathogen-free conditions at 25°C with 12 h light and dark cycles in accordance with current ethical regulations for animal care and use in China. In the experiment, caging bias was taken into account. Instead of feeding them in an independent ventilated cage, they were all placed in the same environment open to the outside.

Remark #5:

please clarify where the NC and EtOH control groups were generated. Were these mice housed with CIA groups, did they undergo similar procedures etc. This point is particularly relevant to the microbiome assessment, as the CIA treatment, handling during disease scoring etc would have increased exposure to bacteria

Reply #5:

Thank reviewer for this insightful suggestion. In order to avoid the influence of environments on mice in different groups, all mice were kept under specific pathogen-free conditions and open to the outside, instead of in independent ventilated cages. Except for drinking water and the variables used to build the model, other treatments were the same for different groups of mice. The previous description in the experimental method did not emphasize these feeding conditions. This experimental process has been added in the revised article (Page 12, Line 472-474).

Remark #6:

is the concentration of acetate used 100mM physiologically relevant to the concentrations observed in the study?

Reply #6:

Thank reviewer for this insightful suggestion. There is no direct relationship between 100 mM acetate used and the concentration of acetate observed. The acetate concentration was observed after giving 10% ethanol in drinking water, and this concentration was observed to study the metabolic process of ethanol in CIA mice. Moreover, the supplementation of 100 mM acetate to CIA+ABX mice was carried out with reference to the literature (Mariño, et al., 2017; Macia, et al., 2015), and it was further determined that it was the acetate produced by the metabolism of the intestinal flora that produced the inhibitory effect on CIA. We are particularly grateful for your suggestion and will further investigate the physiological relevance of the concentration in subsequent experiments.

Remark #7:

6) Figure 2A (flow cytometry) needs justification of gating strategy (e.g FMO or unstained controls) as the gates are not convincing. Also the gates shown do NOT show what is denoted in the legend and text. I.e. Gating shown is not Ly6c+ OR Ly6G+ but CD11b+ only. In addition CD11b+Ly6c+ cells would include monocytes and macrophages, not macrophages as stated in the text

Reply #7:

Thank reviewer for this constructive comment, and we have modified and readjusted

the gating strategy in Figure 2A. The flow cytogram of CD11b+Ly6C+ and CD11b+Ly6G+ was analyzed using the cross-gate quadrant gate method to analyze the percentage of double-positive cells, and all labels were enlarged to be clearly visible. As for CD11b+Ly6c+ cells would include monocytes and macrophages, we have modified this in revision.

General

Remark #8:

line 349 is missing

Reply #8:

Thank reviewer for its careful comment, and we have added this section and completed this sentence (Page 4, line 151-153) as follow:

These data suggest that low-dose ethanol consumption alleviates the clinical manifestations of CIA mice and protects bone from erosion.

Remark #9:

Figure 1C-E is very poor quality

Reply #9:

As reviewer suggested, we have replaced Figure 1C-E with better-quality images.

Remark #10:

Figure 7 is so blurry I can't interpret any of it

Reply #10:

Thank the reviewer for this kind reminder. In order to improve the quality of Figure 7 and display the results more clearly, we have adjusted its resolution and moved some data to Supplementary Figure 3.

Remark #11:

the flow cytometry suggests changes in monocytes and T-cells as well as neutrophils, why were these cells not investigated as associated with the ethanol phenotype?

Reply #11:

Thanks for the reviewer's professional comments. As the reviewer suggested, acetate may have effects on monocytes and T cells, but this study focused on neutrophils to investigate the mechanism of acetate inhibiting NETs for the following reasons. First, neutrophils play an important role in the early stages of RA or CIA, which in turn affects adaptive immune cells (Jaillon et al.,2020). Second, receptors for short-chain fatty acids are mainly expressed on the surface of granulocytes (Carretta et al.,2021), and acetate affects neutrophils through short-chain fatty acid receptors on the surface of granulocytes, thus inhibiting the production of NETs. In this study, we believe that acetate plays an important role in the early stages of CIA related to neutrophils and therefore focus on neutrophils. Furthermore, the effect of ethanol intake on T cells and macrophages will be further explored in our future studies.

Remark #12:

Fig 1K the effect on inflammatory cytokines more significant than Ly6G – suggests this may be the cause for reduced severity of disease rather than via NETs

Reply #12:

Thank reviewer for this insightful comment. Inflammatory cytokines such as IL-1 β , IL-17a, TNF- α and IFN- γ are secreted by T cells, and these inflammatory cytokines in the CIA+EtOH group are significantly less than the CIA group, which indicates that ethanol treated CIA mice have less effector T cell infiltration in the joint. Innate immune cells such as granulocytes and macrophages are activated first in the development of RA. These innate immune cells recruit more immune cell infiltration by secreting chemokines (Legein et al.,2013; Mai et al.,2013), which aggravate joint inflammation, therefore NETs play a more important role in the early stage of RA. In subsequent experiments, we further demonstrated that the intake of EtOH and acetate significantly reduced the formation of NETs, which confirmed that inhibiting the formation of NETs plays a key role in alleviating CIA in mice.

Remark #13:

Line 414 Muribaculaceae was not increased with ethanol only compared to CIA only. Please clarify and justify why ethanol without CIA does not affect Muribaculaceae. 566 repeated

Reply #13:

Thank reviewer for this constructive suggestion, we have discussed this question in the discussion section (Page 10, Line 372-378) as follow:

Muribaculaceae, which belongs to the *Bacteroidae* (Huang et al.,2022), plays an important role in maintaining the integrity of intestinal barrier function and the ability to display anti-inflammatory activity. In this study, although ethanol treatment alone caused the remodeling of intestinal microorganisms, we speculate that it is only when the inflammatory environment and ethanol act at the same time that they can stimulate the abundance of the beneficial bacteria of *Muribaculaceae* to reverse the situation and

return the body to a normal state. Therefore, only when inflammation, it can have a profound impact on *Muribaculaceae*.

Remark #14:

Figure 1: What is total body score? This is not defined, but used to determine incidence. It is unusual to define incidence based on body score and not swelling - swelling appears to have a similar time of onset in figure D.

Reply #14:

We thank the reviewer for this insightful comment. As the reviewer suggested, the usage of body score to define incidence was indeed inappropriate and we have redefined the incidence based on Hind Paw Swelling Score. And the scoring rules for overall body score, hind paw swelling score and arthritis score have been supplemented in the methods (Page 13, line 494-509) as follow:

Arthritis index scoring standard: 2 independent blind observers scored the mouse arthritis index every day, scoring from three aspects of overall body score, hind paw swelling score and arthritis score respectively. Overall body scoring criteria: 0 = no paw swelling; 1 = erythema and swelling in one paw; 2 = erythema and swelling in two paws; 3 = erythema and swelling in three paws; 4 = erythema and swelling in all paws. The mice had a maximum overall body score of 4. Hind paw swelling scoring criteria: 0 = normal paw; 1 = one swelling in the middle of five fingers/toes and wrist/ankle; 2 = two swellings in the middle of five fingers/toes and wrist/ankle; By analogy, the highest score of each paw is 6, and the score of paws swelling number of each mouse is the sum of all paw scores, and the highest hind paw swelling score is 24. Arthritis score: 0 = normal paw; 1 = erythema and mild swelling in the ankle/wrist; 2 = erythema and mild swelling extending from the ankle/wrist to the mid-hind paw/mid-forepaw; 3 = erythema and moderate swelling extending from the digits to the metatarsophalangeal ankles; 4 = erythema and severe swelling of entire paw or ankylosis of the limb. The

arthritis score for each mouse was the sum of all paw scores. The highest arthritis score per mouse is 16.

Remark #15:

Fig 2D is cropped

Reply #15:

Thank reviewer for this careful comment and we have modified figure 2D.

Remark #16:

Fig 6: EtOH increases Netosis, please clarify as this does contradict the overall claims.

Reply #16:

We thank the reviewer for this comment, and apologize for the ambiguity. Fig. 6 shows the results of in vitro experiments. There is no physiological metabolic process and the mechanism of action is different from that in vivo, and ethanol cannot produce acetate to reduce the production of NETs. In Fig. 6E, different doses of ethanol were added to neutrophils with the stimulation of PMA (stimulator of NETs), and it was found that ethanol did not reduce the formation of NETs by the observation of H3cit and MPO. This result also suggests that acetate rather than ethanol itself produces this inhibitory effect. In this article, we mainly focus on ethanol consumption inhibits NETs formation to alleviate RA. Meanwhile, the possible promoting effect of ethanol itself on NETs will investigate further in follow-up experiments.

Reference:

- Szabo, G. & Saha, B. Alcohol's effect on host defense. *Alcohol Res.: Curr. Rev.* 37, 159–170 (2015).
- Lu, B., Solomon, D. H., Costenbader, K. H. & Karlson, E. W. Alcohol consumption and risk of incident rheumatoid arthritis in women: a prospective study. *Arthritis Rheumatol* 66, 1998–2005, doi:10.1002/art.38634 (2014).
- Kallberg, H. et al. Alcohol consumption is associated with decreased risk of rheumatoid arthritis: results from two Scandinavian case-control studies. *Ann Rheum Dis* 68, 222–227, doi:10.1136/ard.2007.086314 (2009).
- Sivaprakasam, S., Prasad, P. D. & Singh, N. Benefits of short-chain fatty acids and their receptors in inflammation and carcinogenesis. *Pharmacol Ther* 164, 144–151, doi:10.1016/j.pharmthera.2016.04.007 (2016).
- Carretta, M. D., Quiroga, J., Lopez, R., Hidalgo, M. A. & Burgos, R. A. Participation of Short-Chain Fatty Acids and Their Receptors in Gut Inflammation and Colon Cancer. *Front Physiol* 12, 662739, doi:10.3389/fphys.2021.662739 (2021).
- Tillack, K., Breiden, P., Martin, R. & Sospedra, M. T lymphocyte priming by neutrophil extracellular traps links innate and adaptive immune responses. *J Immunol* 188, 3150–3159, doi:10.4049/jimmunol.1103414 (2012).
- Papayannopoulos, V. Neutrophil extracellular traps in immunity and disease. *Nat Rev Immunol* 18, 134–147, doi:10.1038/nri.2017.105 (2018).
- Azizov V, Dietel K, Steffen F, Dürholz K, Meidenbauer J, Lucas S, Frech M, Omata Y, Tajik N, Knipfer L, Kolenbrander A, Seubert S, Lapuente D, Sokolova MV, Hofmann J, Tenbusch M, Ramming A, Steffen U, Nimmerjahn F, Linker R, Wirtz S, Herrmann M, Temchura V, Sarter K, Schett G, Zaiss MM. Ethanol consumption inhibits TFH cell responses and the development of autoimmune arthritis. *Nat Commun.* 2020 Apr 24;11(1):1998. doi: 10.1038/s41467-020-15855-z.
- Jonsson IM, Verdrengh M, Brisslert M, Lindblad S, Bokarewa M, Islander U, Carlsten H, Ohlsson C, Nandakumar KS, Holmdahl R, Tarkowski A. Ethanol prevents development of destructive arthritis. *Proc Natl Acad Sci U S A.* 2007 Jan 2;104(1):258–63. doi: 10.1073/pnas.0608620104. Epub 2006 Dec 21.
- GBD 2020 Alcohol Collaborators. Population-level risks of alcohol consumption by amount, geography, age, sex, and year: a systematic analysis for the Global Burden of Disease Study 2020. *Lancet.* 2022 Jul 16;400(10347):185–235. doi: 10.1016/S0140-6736(22)00847-9. Erratum in: *Lancet.* 2022 Jul 30;400(10349):358.
- GBD 2016 Alcohol Collaborators. Alcohol use and burden for 195 countries and territories, 1990–2016: a systematic analysis for the Global Burden of Disease Study 2016. *Lancet.* 2018 Sep 22;392(10152):1015–1035. doi: 10.1016/S0140-6736(18)31310-2. Epub 2018 Aug 23.

Erratum in: *Lancet*. 2018 Sep 29;392(10153):1116. Erratum in: *Lancet*. 2019 Jun 22;393(10190):e44.

Waller, M. B., Murphy, J. M., McBride, W. J., Lumeng, L., & Li, T. K. (1986). Effect of low dose ethanol on spontaneous motor activity in alcohol-preferring and -nonpreferring lines of rats. *Pharmacology, biochemistry, and behavior*, 24(3), 617–623. [https://doi.org/10.1016/0091-3057\(86\)90567-8](https://doi.org/10.1016/0091-3057(86)90567-8)

Kimoto, A., Izu, H., Fu, C., Suidasari, S., & Kato, N. (2017). Effects of low dose of ethanol on the senescence score, brain function and gene expression in senescence-accelerated mice 8 (SAMP8). *Experimental and therapeutic medicine*, 14(2), 1433–1440. <https://doi.org/10.3892/etm.2017.4633>

Bryant, K. G., Singh, B., & Barker, J. M. (2022). Reinforcement History Dependent Effects of Low Dose Ethanol on Reward Motivation in Male and Female Mice. *Frontiers in behavioral neuroscience*, 16, 875890. <https://doi.org/10.3389/fnbeh.2022.875890>

Tizabi, Y., Getachew, B., Ferguson, C. L., Csoka, A. B., Thompson, K. M., Gomez-Paz, A., et al. (2018). Low Vs. high alcohol: central benefits Vs. detriments. *Neurotox. Res.* 34, 860–869. doi: 10.1007/S12640-017-9859-X

Sule G, Abuaita BH, Steffes PA, Fernandes AT, Estes SK, Dobry C, Pandian D, Gudjonsson JE, Kahlenberg JM, O'Riordan MX, Knight JS. Endoplasmic reticulum stress sensor IRE1 α propels neutrophil hyperactivity in lupus. *J Clin Invest*. 2021 Apr 1;131(7):e137866. doi: 10.1172/JCI137866.

Chen Y, Wu Z, Huang S, Wang X, He S, Liu L, Hu Y, Chen L, Chen P, Liu S, He S, Shan B, Zheng L, Duan SZ, Song Z, Jiang L, Wang QA, Gan Z, Song BL, Liu J, Rui L, Shao M, Liu Y. Adipocyte IRE1 α promotes PGC1 α mRNA decay and restrains adaptive thermogenesis. *Nat Metab*. 2022 Sep;4(9):1166-1184. doi: 10.1038/s42255-022-00631-8. Epub 2022 Sep 19.

Brand DD, Latham KA, Rosloniec EF. Collagen-induced arthritis. *Nat Protoc*. 2007;2(5):1269-75. doi: 10.1038/nprot.2007.173.

Miyoshi M, Liu S. Collagen-Induced Arthritis Models. *Methods Mol Biol*. 2018;1868:3-7. doi: 10.1007/978-1-4939-8802-0_1.

Mariño E, Richards JL, McLeod KH, Stanley D, Yap YA, Knight J, McKenzie C, Kranich J, Oliveira AC, Rossello FJ, Krishnamurthy B, Nefzger CM, Macia L, Thorburn A, Baxter AG, Morahan G, Wong LH, Polo JM, Moore RJ, Lockett TJ, Clarke JM, Topping DL, Harrison LC, Mackay CR. Gut microbial metabolites limit the frequency of autoimmune T cells and protect against type 1 diabetes. *Nat Immunol*. 2017 May;18(5):552-562. doi: 10.1038/ni.3713. Epub 2017 Mar 27. Erratum in: *Nat Immunol*. 2017 Jul 19;18(8):951. Erratum in: *Nat Immunol*. 2017 Oct 18;18(11):1271.

Macia L, Tan J, Vieira AT, Leach K, Stanley D, Luong S, Maruya M, Ian McKenzie C, Hijikata A, Wong C, Binge L, Thorburn AN, Chevalier N, Ang C, Marino E, Robert R, Offermanns S, Teixeira MM, Moore RJ, Flavell RA, Fagarasan S, Mackay CR. Metabolite-sensing receptors

GPR43 and GPR109A facilitate dietary fibre-induced gut homeostasis through regulation of the inflammasome. *Nat Commun.* 2015 Apr 1;6:6734. doi: 10.1038/ncomms7734.

Jaillon, S. et al. Neutrophil diversity and plasticity in tumour progression and therapy. *Nat Rev Cancer* 20, 485-503, doi:10.1038/s41568-020-0281-y (2020).

Carretta, M. D., Quiroga, J., Lopez, R., Hidalgo, M. A. & Burgos, R. A. Participation of Short-Chain Fatty Acids and Their Receptors in Gut Inflammation and Colon Cancer. *Front Physiol* 12, 662739, doi:10.3389/fphys.2021.662739 (2021).

Carretta MD, Quiroga J, López R, Hidalgo MA, Burgos RA. Participation of Short-Chain Fatty Acids and Their Receptors in Gut Inflammation and Colon Cancer. *Front Physiol.* 2021 Apr 8;12:662739. doi: 10.3389/fphys.2021.662739.

Legein B, Temmerman L, Biessen EA, Lutgens E. Inflammation and immune system interactions in atherosclerosis. *Cell Mol Life Sci.* 2013 Oct;70(20):3847-69. doi: 10.1007/s00018-013-1289-1. Epub 2013 Feb 21.

Mai J, Virtue A, Shen J, Wang H, Yang XF. An evolving new paradigm: endothelial cells--conditional innate immune cells. *J Hematol Oncol.* 2013 Aug 22;6:61. doi: 10.1186/1756-8722-6-61.

Huang J, Liu D, Wang Y, Liu L, Li J, Yuan J, Jiang Z, Jiang Z, Hsiao WW, Liu H, Khan I, Xie Y, Wu J, Xie Y, Zhang Y, Fu Y, Liao J, Wang W, Lai H, Shi A, Cai J, Luo L, Li R, Yao X, Fan X, Wu Q, Liu Z, Yan P, Lu J, Yang M, Wang L, Cao Y, Wei H, Leung EL. Ginseng polysaccharides alter the gut microbiota and kynurenine/tryptophan ratio, potentiating the antitumour effect of antiprogrammed cell death 1/programmed cell death ligand 1 (anti-PD-1/PD-L1) immunotherapy. *Gut.* 2022 Apr;71(4):734-745. doi: 10.1136/gutjnl-2020-321031. Epub 2021 May 18.

REVIEWERS' COMMENTS:

Reviewer #1 (Remarks to the Author):

The authors addressed all points of concern by adding more background knowledge and details from previously published studies in this field. Images are added as high-resolution images and are all clearly visible. No additional measurements or experiments were performed.

I have minor suggestions:

The summarizing paragraph in the introduction section needs to be revised, since the phrasing is not clear. Same with the explanation for IRE1 activation, line 174 ff and the summary line 375 ff.

Colors in graphs 1c-d could be matched with the bar charts e and following. Legend for 1d is not matching axes labeling.

Color matching for figure 4 and 5 would be helpful and also throughout the manuscript.

Axes labeling in figure 5 - relative mRNA level - needs to be corrected.

A complete gating strategy should be included for flow cytometry experiments. Ly6C is also expressed by neutrophils, but it seems to be not considered in the gating.

Reviewer #2 (Remarks to the Author):

This article by Jin et al is an interesting study that aims to elucidate the links between ethanol consumption and arthritis. I am happy that after initial review my comments have been well addressed and the authors have diligently corrected any issues.

Point-by-point responses to Reviewers' comments

Reviewer #1:

The authors addressed all points of concern by adding more background knowledge and details from previously published studies in this field. Images are added as high-resolution images and are all clearly visible. No additional measurements or experiments were performed.

I have minor suggestions:

Remark #1:

The summarizing paragraph in the introduction section needs to be revised, since the phrasing is not clear. Same with the explanation for IRE1 activation, line 174 ff and the summary line 375 ff.

Reply #1:

Thank you for the helpful suggestion. We agree with this suggestion and have corrected the summarizing paragraph in the introduction section (Page 3, Line 93-105) as follow:

In this study, we first detected arthritis indicators in a low-dose alcohol treatment model of CIA mice and found that ethanol treatment can alleviate the progression of CIA mice. Using a range of Tissue staining observation and biological detection technology, we elucidated the ethanol treatment inhibits the formation of NETs in the joints. Then, microbiomic and metabolomic analyses revealed that Muribaculaceae was predominant in the gut microbiota of mice after ethanol treatment, and the levels of microbiota metabolite acetate were increased. Meanwhile, acetate reduced ERS in neutrophils and inhibited the formation of NETs. Furthermore, exogenous acetate reversed CIA mice with exacerbated gut microbial disruption, further confirming that the effect of gut microbial metabolite acetate on neutrophils in vivo is crucial for the immune regulation. This work emphasizes the pivotal role of acetate during RA intervention, which may offer alternative avenues to replicate or induce the joint-protective benefits of ethanol without associated detrimental effects.

Activation of IER1 can be detected by the expression of p-IRE (Sule et al., 2021; Chen et al., 2022), therefore we modified the activation of IRE to the phosphorylation of IRE (Page 5, Line 175-178

and Page 5, Line 189-192) as follow:

When ERS occurs, IRE1 is activated through autophosphorylation, showing endonuclease activity, cutting the downstream molecule (XBP1) base of IRE1 signaling pathway, and generating an active spliceosome XBP1s to regulate the downstream.

.....which indicates that there is activation of ERS response sensor IRE1 phosphorylation and increased production of NETs in RA. In addition, immunofluorescence staining also confirmed that there was over-production of p-IRE1 in the neutrophils of synovial tissues of RA patients (Fig. 3d).

Remark #2:

Colors in graphs 1c-d could be matched with the bar charts e and following. Legend for 1d is not matching axes labeling.

Color matching for figure 4 and 5 would be helpful and also throughout the manuscript.

Reply #2:

Thank you for the constructive opinions. We have matched the colors of different groups in the graph. In Figure 1 to Figure 5, purple represents the NC group, blue represents the CIA group, green represents the EtOH group, and orange represents the EtOH +CIA group. All colors have been uniformed.

The 1D legend has been modified compared with the coordinate axis (Page 24, Line 874-875) as follow:

(D) Arthritis severity during CIA, as assessed by arthritis score, overall body score, and hind paw swelling score.

D

Remark #3:

Axes labeling in figure 5 - relative mRNA level - needs to be corrected.

Reply #3:

Thanks for your careful suggestion, we have corrected it.

C

Remark #4:

A complete gating strategy should be included for flow cytometry experiments. Ly6C is also expressed by neutrophils, but it seems to be not considered in the gating.

Reply #4:

Thank you for the thoughtful suggestion. We have adjusted the display of flow cytometry results; the gating strategy has been supplemented in Figure 2A.

Regarding the marker of flow cytometry to detect the neutrophils and macrophage, thanks for your professional suggestion again. We designed the experiment with reference to the work of Shin AE. et al. (Gastroenterology. 2023) and Huang et al. (Hepatology. 2022), who defined $CD11b^+Ly6G^+$ and $CD11b^+Ly6C^+$ as neutrophils and macrophages, respectively. Low dose Ly6C is also expressed by neutrophils, it is an oversight in our work. $CD11b(+)$ Ly6G(hi) neutrophils and $CD11b(+)$ Ly6C(hi) monocytes, more are widely used. After learning this, we attempted to reanalyze the flow cytometry results. However, since the cells analyzed were derived from digested mouse paws, different cells are not easily displayed in groups, so the distinction between high and low expression of Ly6G and Ly6C is not significant. Thanks again to the reviewer for their valuable comments, we will discuss this further in subsequent experiments.

Reviewer #2:

This article by Jin et al is an interesting study that aims to elucidate the links between ethanol consumption and arthritis. I am happy that after initial review my comments have been well addressed and the authors have diligently corrected any issues.

Reply:

Thank you very much again for this feedback and for your valuable critique of our initial manuscript version.

Reference:

Sule G, Abuaita BH, Steffes PA, Fernandes AT, Estes SK, Dobry C, Pandian D, Gudjonsson JE, Kahlenberg JM, O'Riordan MX, Knight JS. Endoplasmic reticulum stress sensor IRE1 α propels neutrophil hyperactivity in lupus. *J Clin Invest.* 2021 Apr 1;131(7):e137866. doi: 10.1172/JCI137866.

Chen Y, Wu Z, Huang S, Wang X, He S, Liu L, Hu Y, Chen L, Chen P, Liu S, He S, Shan B, Zheng L, Duan SZ, Song Z, Jiang L, Wang QA, Gan Z, Song BL, Liu J, Rui L, Shao M, Liu Y. Adipocyte IRE1 α promotes PGC1 α mRNA decay and restrains adaptive thermogenesis. *Nat Metab.* 2022 Sep;4(9):1166-1184. doi: 10.1038/s42255-022-00631-8. Epub 2022 Sep 19.

Shin AE, Tesfagiorgis Y, Larsen F, Derouet M, Zeng PYF, Good HJ, Zhang L, Rubinstein MR, Han YW, Kerfoot SM, Nichols AC, Hayakawa Y, Howlett CJ, Wang TC, Asfaha S. F4/80+Ly6Chigh Macrophages Lead to Cell Plasticity and Cancer Initiation in Colitis. *Gastroenterology.* 2023 Apr;164(4):593-609.e13. doi: 10.1053/j.gastro.2023.01.002. Epub 2023 Jan 10. PMID: 36634827; PMCID: PMC10038892.

Huang M, Jiao J, Cai H, et al. C-C motif chemokine ligand 5 confines liver regeneration by down-regulating reparative macrophage-derived hepatocyte growth factor in a forkhead box O 3a-dependent manner. *Hepatology.* 2022;76(6):1706-1722.